# Microfluidic platform accelerates tissue processing into single cells for molecular analysis and primary culture models

Jeremy A. Lombardo[1], Marzieh Aliaghaei[2], Quy H. Nguyen[3], Kai Kessenbrock [3,4] & Jered B. Haun [1,2,4,5,6✉]

Tissues are complex mixtures of different cell subtypes, and this diversity is increasingly characterized using high-throughput single cell analysis methods. However, these efforts are hindered, as tissues must first be dissociated into single cell suspensions using methods that are often inefficient, labor-intensive, highly variable, and potentially biased towards certain cell subtypes. Here, we present a microfluidic platform consisting of three tissue processing technologies that combine tissue digestion, disaggregation, and filtration. The platform is evaluated using a diverse array of tissues. For kidney and mammary tumor, microfluidic processing produces 2.5-fold more single cells. Single cell RNA sequencing further reveals that endothelial cells, fibroblasts, and basal epithelium are enriched without affecting stress response. For liver and heart, processing time is dramatically reduced. We also demonstrate that recovery of cells from the system at periodic intervals during processing increases hepatocyte and cardiomyocyte numbers, as well as increases reproducibility from batch-to-batch for all tissues.

[1] Department of Biomedical Engineering, University of California, Irvine, Irvine, CA, USA. [2] Department of Chemical and Biomolecular Engineering, University of California, Irvine, Irvine, CA, USA. [3] Department of Biological Chemistry, School of Medicine, University of California, Irvine, Irvine, CA, USA. [4] Chao Family Comprehensive Cancer Center, University of California, Irvine, Irvine, CA, USA. [5] Department of Materials Science and Engineering, University of California, Irvine, Irvine, CA, USA. [6] Center for Advanced Design and Manufacturing of Integrated Microfluidics, University of California, Irvine, Irvine, CA, USA. ✉email: jered.haun@uci.edu

Tissues are highly complex ecosystems containing a diverse array of cell subtypes. Significant variation can also arise within a given subtype due to differences in activation state, genetic mutations, epigenetic distinctions, stochastic events, and microenvironmental factors[1,2]. This has led to a rapid growth in studies attempting to capture cellular heterogeneity, and thereby gain a better understanding of tissue and organ development, normal function, and disease pathogenesis[3–9]. For example, in the context of cancer, intratumor heterogeneity is a key indicator of disease progression, metastasis, and the development of drug resistance[10–14]. High-throughput single-cell analysis methods such as flow cytometry, mass cytometry, and single-cell RNA sequencing (scRNA-seq) are ideal for identifying single cells in a comprehensive manner based on molecular information[15,16], and these methods have already begun to transform our understanding of complex tissues by enabling identification of previously unknown cell types and states[8,17–19]. However, a critical barrier to these efforts is the need to first process tissues into a suspension of single cells. Current methods involve mincing, digestion, disaggregation, and filtering that are labor-intensive, time consuming, inefficient, and highly variable[20,21]. Thus, new approaches and technologies are critically needed to ensure reliability and wide-spread adoption of single-cell analysis methods for tissues. This would be particularly important for translating single-cell diagnostics to human specimens in clinical settings. Moreover, improved tissue dissociation would make it faster and easier to extract primary cells for ex vivo drug screening, engineered tissue constructs, and stem/progenitor cell therapies[22–25]. Patient-derived organ-on-a-chip models, which seek to recapitulate complex native tissues for personalized drug testing, are a particularly exciting future direction that could be enabled by improved tissue dissociation[22,26–30].

scRNA-seq has recently emerged as a powerful and widely adaptable analysis technique that provides the full transcriptome of individual cells. This has enabled comprehensive cell reference maps, or atlases, to be generated for normal and diseased tissues, as well as identification of previously unknown cell subtypes or functional states[31,32]. For example, an atlas recently generated for normal murine kidney uncovered a new collecting duct (CD) cell with a transitional phenotype and an unexpected level of cellular plasticity[4]. Moreover, an atlas of primary human breast epithelium linked distinct epithelial cell populations to known breast cancer subtypes, suggesting that these subtypes may develop from different cells of origin[3]. For melanoma, scRNA-seq was used to identify three transcriptionally distinct states, one of which was drug sensitive, and further demonstrated that drug resistance could be delayed using computationally optimized therapy schedules[33]. While scRNA-seq is clearly a powerful diagnostic modality, the process of breaking down the tissue into single cells can introduce confounding factors that may negatively influence data quality and reliability. One factor is the lack of standardization, which can lead to substantial variation across different research groups and tissue types. Another significant concern is that incomplete breakdown could bias results toward cell types that are easier to liberate. A recent study utilizing single-nuclei RNA sequencing with murine kidney samples found that endothelial cells and mesangial cells (MC) were underrepresented in scRNA-seq data[34]. Finally, lengthy enzymatic digestion times have been shown to alter transcriptomic signatures and generate stress responses that interfere with cell classification[35–39]. Addressing these concerns would help propel the exciting field of scRNA-seq into the future for tissue atlasing and disease diagnostics.

Microfluidic technologies have advanced the fields of biology and medicine by miniaturizing devices to the scale of cellular samples and enabling precise sample manipulation[40–44]. Most of this work has focused on manipulating and analyzing single cells[44–48]. Only a small number of studies have addressed tissue processing, and even fewer have focused on breaking down tissue into smaller constituents[49–51]. We previously developed a microfluidic device that specifically focused on breaking down cellular aggregates into single cells[52,53]. This dissociation device contained a network of branching channels that progressively decreased in size down to ~100 μm, and contained repeated expansions and constrictions to break down aggregates using shear forces. We then developed a device for on-chip tissue digestion using the combination of shear forces and proteolytic enzymes[54]. Finally, we developed a filter device containing nylon mesh membranes that removed large tissue fragments, while also dissociating smaller cell aggregates and clusters[55]. The microfluidic digestion, dissociation, and filter devices each enhanced single cell recovery when operated independently. To date, however, we have not combined these technologies to maximize performance and execute a complete tissue processing workflow on-chip. Moreover, we have not validated microfluidically processed cell suspensions using scRNA-seq.

In this work, we present a microfluidic platform comprised of three different tissue-processing technologies that enhances the breakdown of tissue and produces single-cell suspensions that are immediately ready for downstream single-cell analysis or other use. First, we design a digestion device that can be loaded with minced tissue and operated with minimal user interaction. Next, we integrate the dissociation and filter technologies into a single unit, and optimize the two-device platform using murine kidney to produce single cells more quickly and in higher numbers than traditional methods. Using the optimized protocol, we evaluate different tissue types using two single-cell analysis methods. For murine kidney and breast tumor tissues, microfluidic processing can produce >2-fold more epithelial cells and leukocytes, and >5-fold more endothelial cells. Using scRNA-seq, we show that device processed samples are highly enriched for endothelial cells, fibroblasts, and basal epithelium. We also demonstrate that stress responses are not induced in any cell type, and can even be reduced if shorter processing times are employed. For murine liver and heart, significant single cell numbers are obtained after only 15 min, and even as short as 1 min. Interestingly, we find that substantially more hepatocytes and cardiomyocytes are obtained if sample is recovered at discrete intervals, most likely because these cell types are sensitive to shear forces. Importantly, the microfluidic platform can significantly shorten processing time or enhance single cell recovery for all tissue types studied, and in some cases accomplish both, while increasing batch-to-batch reproducibility and maintaining viability. Furthermore, the entire tissue processing workflow is performed in an automated fashion. Thus, our microfluidic platform holds exciting potential to advance diverse applications that require the liberation of single cells from tissues.

## Results

We designed a digestion device that would not require manual device assembly. Instead, minced tissue is loaded through a port at the top of the device, which can then be sealed using a cap or stopcock. Scalpel mincing of tissue into ~1 mm³ pieces is ubiquitous, and therefore this format will be compatible with a wide array of tissue types and dissociation protocols. The full design layout of the minced tissue digestion device is shown in Fig. 1a, including the loading port, a chamber that retains the tissue in place, and fluidic channels that administer fluid shear forces and deliver proteolytic enzymes. These features were arranged across six layers of hard plastic, including two fluidic channel layers, two "via" layers, a top end cap with hose barbs and loading port, and a

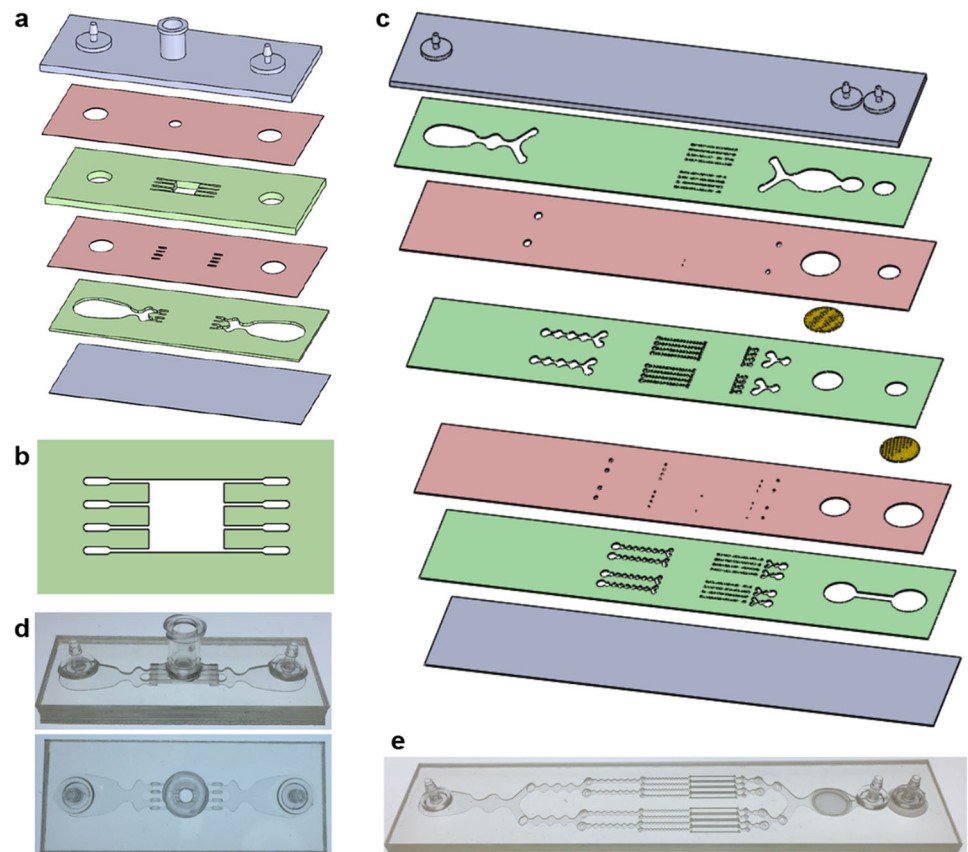

**Fig. 1 Microfluidic tissue processing platform. a** Schematic of the minced tissue digestion device. Design includes six total layers, including two fluidic layers (green), 2 via layers (red), and the top and bottom end caps (gray). Tissue is loaded through the luer port and into the tissue chamber. **b** Schematic of the tissue chamber. Fluidic channels direct hydrodynamic shear forces and proteolytic enzymes, while also retaining minced tissue pieces in the chamber. **c** Schematic of the integrated dissociation/filter device. Tissue fragments and cell aggregates from the digestion device will be further broken down by hydrodynamic shear forces and nylon mesh filters. **d** Pictures of the fabricated minced digestion device. **e** Picture of the fabricated dissociation/filter device.

bottom end cap. The tissue chamber is in the uppermost fluidic layer, directly beneath the loading port and a 2.5 mm diameter via, and a detailed schematic is shown in Fig. 1b. We employed a square geometry, with 5 mm length and width, to allow tissue to be evenly distributed during loading. Chamber height was 1.5 mm, slightly larger than minced tissue, to prevent clogging during sample loading and device operation. Fluidic channels were placed upstream and downstream of the tissue chamber, and in both cases, we employed four channels that were 250 μm wide. The symmetric channel design was chosen for the minced format because there is a greater emphasis on prevention of clogging. We also extended channel length to 4 mm to prevent larger tissue pieces from squeezing all the way through the channels, but flared the end to make it easier to connect with the underlying via layer.

The dissociation and filter devices will process tissue fragments and cell aggregates that are small enough to leave the tissue chamber of the digestion device. This includes disaggregation via shear forces generated within the branching channel array and via physical interaction with nylon mesh filters[52,55]. For this work, we have integrated the dissociation and filter devices into a single unit to minimize holdup volume and simplify operation. The original designs were both arranged across seven layers, and thus were directly integrated, as shown in Fig. 1c.

The minced digestion and integrated dissociation/filter devices were fabricated using a commercial laminate process, with channel features laser micro-machined into hard plastic (PMMA or poly-ethylene terephthalate (PET)). All layers and other components

were then aligned and bonded using pressure sensitive adhesive. Photographs of the fabricated devices are shown in Fig. 1d, e.

We first evaluated the minced digestion device using adult murine kidney samples. The kidney is a complex organ composed of anatomically and functionally distinct structures, and adult kidney tissue has a dense extracellular matrix that is challenging to dissociate into single cells[52,55,56]. Freshly dissected kidneys were minced using a scalpel to ~1 mm³ pieces and loaded into the minced digestion device through the luer port. The device and tubing were then primed with PBS containing 0.25% type I collagenase, the luer input port was sealed using a stopcock, and fluid was recirculated through the device using a peristaltic pump. We initially tested flow rates of 10 and 20 mL/min, which were used in previous work[52,54,55]. After 15 or 60 min of recirculation, sample was collected, washed, and genomic DNA (gDNA) was extracted to assess total cell recovery. A control was minced and gDNA was directly extracted to provide an upper recovery limit. At 10 mL/min, gDNA was ~15% and 60% of the control after 15 and 60 min, respectively (Fig. 2a). Increasing flow rate to 20 mL/min improved results to ~40% and 85%, respectively. Images of the tissue chamber were captured at the end of each experiment, and representative results are shown in Fig. 2b. We consistently observed tissue remaining in the chamber or adjacent channels at 10 mL/min, corroborating low gDNA recovery results. After 60 min at 20 mL/min, only a small amount of tissue was found within channels/vias, which helps explain why gDNA recovery was slightly lower than the control. Another possibility is that

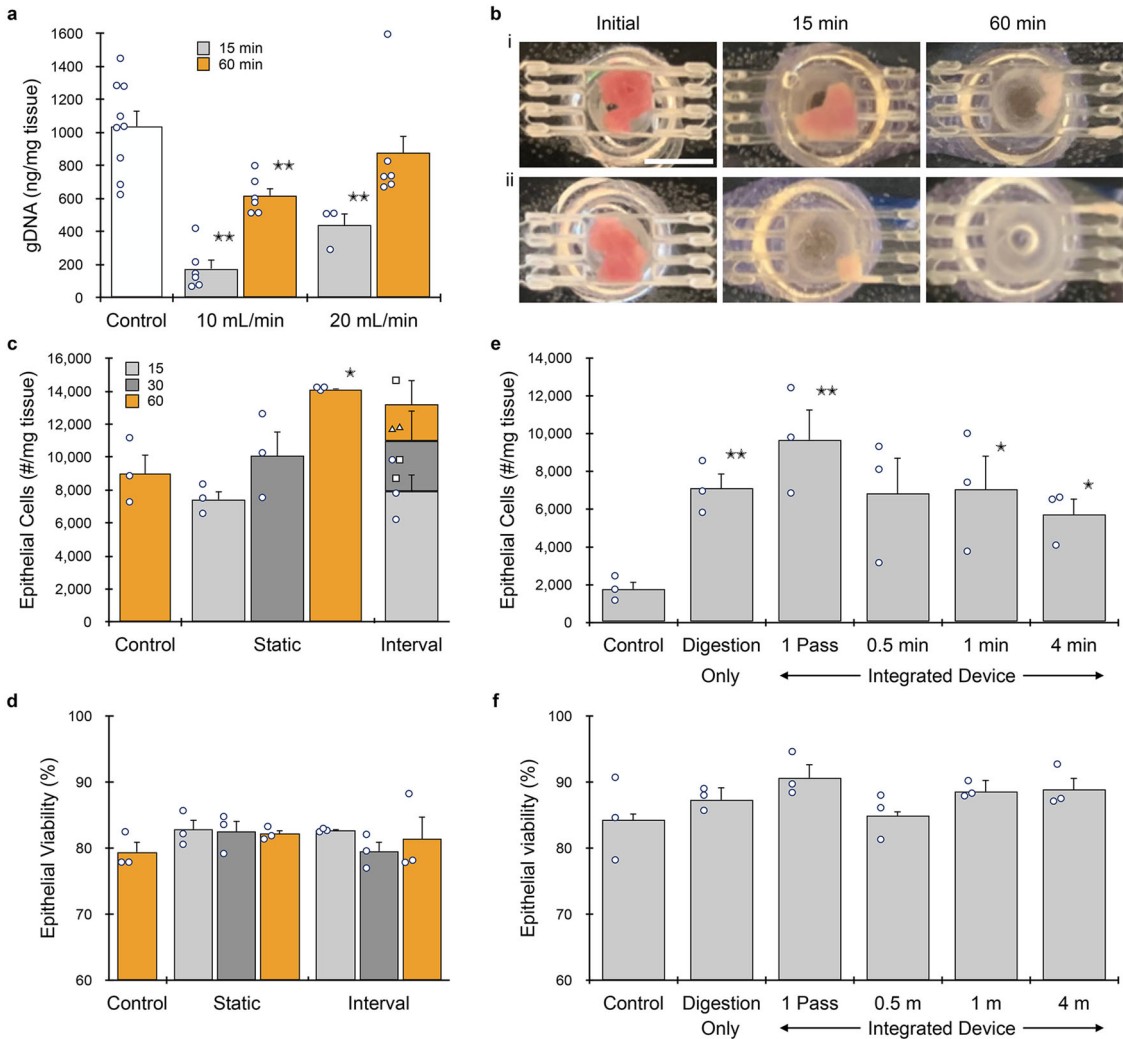

**Fig. 2 Device optimization using murine kidney. a** Kidneys (*n* = 3 to 9 independent samples) were harvested, minced, and processed using the minced digestion device at 10 or 20 mL/min flow rate for 15 or 60 min, and total genomic DNA (gDNA) was quantified. gDNA was extracted directly from the control, and thus this represents maximum recovery. Results at 20 mL/min flow rate were superior, particularly at the shorter time point. **b** Pictures of tissue within the minced digestion device chamber before and after 15 or 60 min of processing at 10 (i) or 20 (ii) mL/min flow rate. Significant tissue remained at 10 mL/min, while tissue was largely absent at 20 mL/min. **c** Single EpCAM+ epithelial cells were quantified by flow cytometry after samples (*n* = 3) were processed with the minced digestion device for 15, 30, or 60 min. We also evaluated recovery of sample at different time intervals, with more collagenase added to continue processing of remaining tissue. **d** Epithelial cell viability was ~80% for all control and device conditions (*n* = 3). **e** Samples (*n* = 3) were processed with the integrated dissociation/filter device following 15 min of digestion device treatment. A single pass through the integrated device produced optimal results. **f** Epithelial cell viability was at ~85–90% for all conditions (*n* = 3). Data are presented as mean values ± SEM from at least three independent experiments. Circles indicate values for experimental replicates. For the stacked plot, experimental replicates are indicated by circles at 15 min, squares at 30 min, and triangles at 60 min. Two-sided *T* test was used for statistical testing. Stars indicate *p* < 0.05 and double stars indicate *p* < 0.01 relative to the control at the same digestion time. *p* values for all comparisons are presented in the Source Data file.

cells were damaged or destroyed during recirculation. To address this concern, we recirculated MCF-7 breast cancer cells through the system and assessed cell number and viability (see Supplementary Information, Supplementary Fig. S1). We observed that cell recovery decreased by ~10% after recirculating through the digestion device, regardless of flow rate or time. Moreover, results were similar after recirculating through the peristaltic pump alone, and cell viability remained high for all conditions tested. This confirms that sample loss was most likely related to holdup within the system or transfer steps, and not damage. Since 20 mL/min was more effective at clearing the tissue chamber and isolating gDNA, this flow rate was used for the remainder of the study.

Next, we analyzed single cells using flow cytometry. Cell suspensions were labeled using a panel of antibodies and fluorescent

probes specific for EpCAM (epithelial cells), TER119 (red blood cells), CD45 (leukocytes), and 7-AAD (live/dead), as listed in Table 1. We found that single epithelial cell numbers increased with processing time, from 15 to 60 min, producing up to ~14,000 cells/mg tissue (Fig. 2c). This represents a 1.5-fold increase relative to the control, which was digested for 60 min under constant agitation, followed by repeated pipetting and vortexing to replicate standard tissue dissociation protocols. We also note that after only 15 min in the digestion device, epithelial cells were statistically similar to the control digested for four-fold longer time. We also investigated an interval operation format, which involved processing for short time periods, eluting the cell suspension, replacing collagenase in the digestion device, and continuing recirculation. We observed that epithelial cell numbers accumulated through each time point of interval operation in a

**Table 1 Flow cytometry probe panels.**

| Assay | Antibody | | Fluorophore | | | | | Positive cells |
|---|---|---|---|---|---|---|---|---|
| | Clone | Dilution (µg/mL) | Kidney (initial) | Kidney (final) | Tumor | Liver | Heart | |
| EpCAM | G8.8 | 7 | PE | PE | PE | N/A | N/A | Epithelial cells |
| TER-119 | TER-119 | 5 | AF647 | AF647 | AF647 | AF647 | AF647 | Red blood cells |
| CD45 | 30-F11 | 5 (AF488) or 12.5 (BV510) | AF488 | BV510 | BV510 | BV510 | BV510 | Leukocytes |
| Viability | N/A | 3.33 (7-AAD) or 1:1000 (ZV) | 7-AAD | 7-AAD | 7-AAD | 7-AAD | Zombie Violet | Dead cells |
| CD31 | MEC13.3 | 8 | N/A | AF488 | AF488 | AF488 | AF488 | Endothelial cells |
| ASGPR1 | 8D7 | 10 | N/A | N/A | N/A | PE | N/A | Hepatocytes |
| Troponin T | REA400 | 0.15 | N/A | N/A | N/A | N/A | PE | Cardiomyocytes |

comparable manner to static operation. This demonstrates that interval collection does not compromise results, and suggests that epithelial cells can withstand long-term recirculation. Epithelial cell viability was ~80% for all control and device conditions, further confirming that device processing did not adversely affect cells (Fig. 2d). Results in terms of cell number and viability were similar for leukocytes (see Supplementary Information, Supplementary Fig. S2a, b).

We then investigated whether the integrated dissociation/filter device can further enhance single cell yield following the digestion device. We performed initial tests using the MCF-7 model, and found that recirculation at 20 mL/min, even for short periods of time, resulted in low viability (see Supplementary Information, Supplementary Fig. S1c,f). At 10 mL/min, single cells increased by ~20% after 30 s of recirculation, with no change in viability, which is similar to previous work using a syringe pump[53]. Longer recirculation times enhanced single cell numbers but decreased viability. Thus, we selected to evaluate short recirculation times at 10 mL/min using minced kidney that had been processed using the digestion device for 15 min. As a final step, sample was passed through the nylon mesh membranes in the filter component at 10 mL/min. Single epithelial cell recovery numbers are presented in Fig. 2e. The digestion device produced four-fold more single cells than the control that was also digested for 15 min. A single pass through the integrated device increased single epithelial cells by ~40% compared to digestion alone, which was ~5.5-fold greater than the control. Recirculation through the branching channel array produced fewer cells than the single pass. Epithelial cell viability was ~85–90% for all conditions (Fig. 2f). Similar results were observed for leukocytes (see Supplementary Information, Supplementary Fig. S2c,e). Based on these results, we selected a single pass through the integrated dissociation/filtration device for all work with kidney. We also note that the integrated device obviates the need for a cell straining step prior to flow cytometry.

We further evaluated kidney under different digestion times using the full microfluidic platform. We also added endothelial cells (via CD31, Table 1) to the flow cytometry panel, since they are difficult to isolate using traditional dissociation methods[34]. Minced tissue was loaded into the digestion device and processed under static (15 or 60 min) or interval (1, 15, and 60 min) formats, and then passed through the integrated dissociation/filter device one time. Controls were minced, digested (15 or 60 min), disaggregated by vortexing/pipetting, and filtered using a cell strainer. Results for epithelial cells are presented in Fig. 3a, and are generally similar to optimization studies (Fig. 2c), although epithelial cells increased to ~20,000/mg tissue. This was ~40% higher than the optimization study due to the integrated dissociation/filter, and overall more than double the 60 min control. Surprisingly, the 1 min interval produced ~1500 epithelial cells/mg, which was similar to the 15 min control. This time point was chosen primarily to eliminate erythrocytes (see Supplementary

Information, Supplementary Fig. S3). Device processing was even more effective for endothelial cells (Fig. 3b), which exceeded the 60 min control by >5-fold. Findings for leukocytes (Fig. 3c) were generally similar to epithelial cells. We note a slight decrease in total cell recovery for the interval format relative to the 60 min static condition for all cell types, although this was not statistically significant. This modest decrease may have been due to sample loss during transfer and/or priming steps. Alternatively, cell clusters may have eluted in the early intervals, which would have otherwise been broken down if they remained within the digestion device. Population distributions for each cell type and processing condition are shown in Fig. 3d. Relative to the 60 min control, endothelial cells were enriched for all device conditions except the 1 min interval. Leukocytes were present at similar levels except in the 15 min control, where they were underrepresented. Interestingly, batch-to-batch reproducibility, as measured by the coefficient of variance (see Supplementary Information, Supplementary Table S1), decreased with processing time for each condition, and was lowest for the microfluidic system using intervals. Viability for all three cell types after device processing was similar to or exceeded controls (see Supplementary Information, Supplementary Fig. S4).

Next we performed scRNA-seq, which has been used to catalog the diverse cell types residing within murine kidney and to create atlases[4,56–59]. Kidney tissue was processed using the device platform and collected at 15 and 60 min intervals, and we also evaluated a 60 min control. Live single cells were isolated from debris and dead cells using fluorescence-activated cell sorting (FACS), loaded onto a droplet-enabled 10X Chromium platform, and 34,034 cells were sequenced at an average depth of ~60,000 reads/cell. scRNA-seq quality metrics are shown in Supplementary Information, Supplementary Table S2, and were comparable across the conditions. After filtering, we used Seurat to identify (Fig. 4a) and annotate (see Supplementary Information, Supplementary Fig. S5) seven cell clusters[60]. This included two clusters of proximal tubules (convoluted, or S1, and straight, or S2–S3), endothelial cells, macrophages, B lymphocytes, and T lymphocytes. The final cluster was heterogenous, and included cells from the distal convoluted tubule (DCT), Loop of Henle (LOH), and CD, as well as MC. All seven clusters were represented in control and device conditions (see Supplementary Information, Supplementary Fig S6a). The relative number of cells in each cluster is shown in Fig. 4b. Proximal tubules were the predominant cell population, representing ~53% of the control, which closely matched a recently published mouse kidney atlas[4]. Proximal tubules were further enriched in the 15 min device condition, comprising ~86% of the cell suspension. The other cell populations were underrepresented relative to the control, most by ~2-fold, but reaching as high as 8-fold for macrophages. However, it is unclear whether this was caused by diminished recovery or simply dilution by proximal tubules. The 60 min device interval only contained ~29% proximal tubules, but we surmise that most

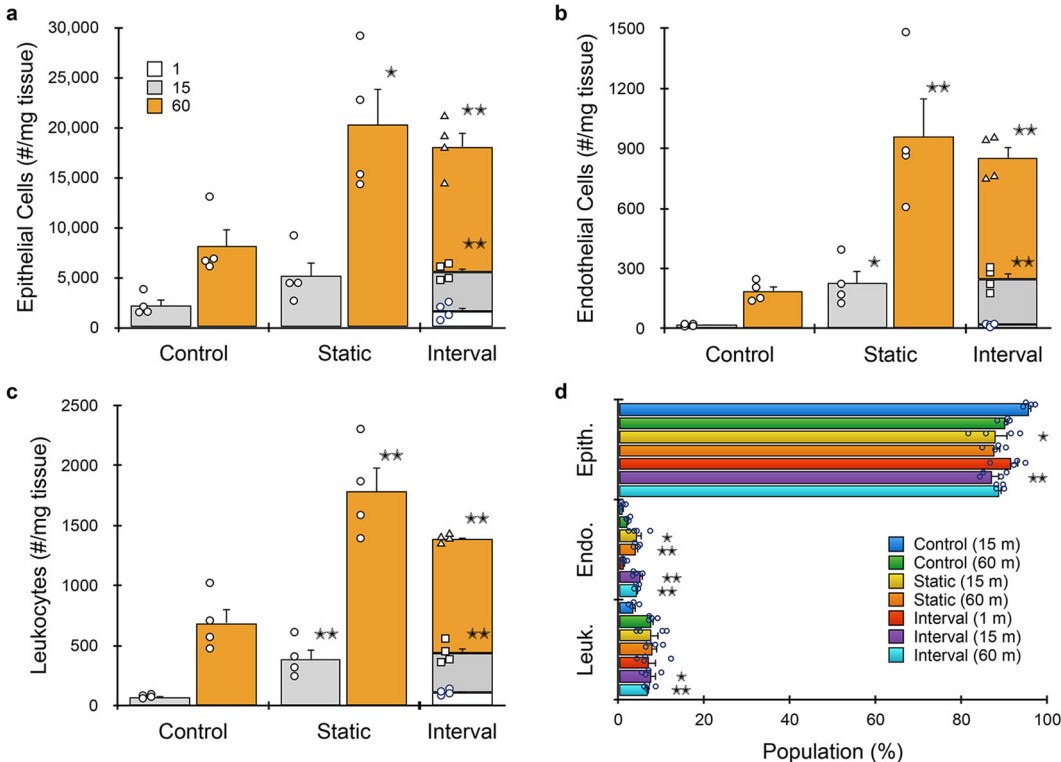

**Fig. 3 Microfluidic platform results for murine kidney.** Kidneys ($n = 4$ independent samples) were harvested, minced, processed with the digestion device for 15 or 60 min, passed through the integrated dissociation/filtration device one time, and resulting cell suspensions were analyzed using flow cytometry. We also evaluated interval recovery at 1, 15, and 60 min time points from the same tissue sample. Controls were minced, digested for either 15 or 60 min, pipetted/vortexed, and passed through a cell strainer. **a** Single EpCAM+ epithelial cells increased by 2.5-fold with microfluidic processing. Interval results were comparable to static, and the 1 min interval produced comparable cell numbers to the 15 min control. Trends were similar for **b** endothelial cells and **c** leukocytes. Microfluidic processing was particularly effective for endothelial cells, yielding >5-fold more cells than the control at 60 min. **d** Population distributions obtained for each processing condition. Endothelial cells were enriched for all device conditions except the 1 min interval relative to controls. Data are presented as mean values ± SEM from at least three independent experiments. Circles indicate values for experimental replicates. For stacked plots, experimental replicates are indicated by circles at 15 min, squares at 30 min, and triangles at 60 min. Two-sided $T$ test was used for statistical testing. Stars indicate $p < 0.05$ and double stars indicate $p < 0.01$ relative to the control at the same digestion time. $p$ values for all comparisons are presented in the Source Data file.

had already been removed in the 15 min interval. Endothelial cells were clearly enriched at 60 min, increasing to ~25% of the suspension, while remaining cell types remained close to control values. Similar trends were observed within the DCT, LOH, DC, and MC sub-clusters (see Supplementary Information, Supplementary Figs. S6b, c). To compare population percentages obtained from scRNA-seq (Fig. 4b) and flow cytometry (Fig. 3d), consideration must be given to which cell populations were likely to express each marker. *CD45* and *CD31* gene expression was well correlated with the appropriate clusters (see Supplementary Information, Supplementary Fig. S7). For *EpCAM*, DCT and CD cells have been shown to express at high levels, while proximal tubules and LOH cells ranged from low to undetectable[61]. Inspection of sequencing results indicated that *EpCAM* was highly expressed by at least a subset of the DCT, LOH, CD, and MC sub-clusters (see Supplementary Information, Supplementary Fig. S7). Interestingly, proximal tubules were predominantly *EpCAM*-negative, but this could be explained by low basal expression and/or a potential secondary factor such as low protein turnover. We used the brightest fluorophore phycoerithrin (PE), to stain EpCAM to help discern low level expression, but it is possible that some cell proximal tubules remained undetectable. Assuming all proximal tubule, DCT, LOH, CD, and MC clusters were EpCAM+, we calculated population percentages of ~62, 88, and 40% for the control, 15 min device, and 60 min device

conditions, respectively. This is directly in line with flow cytometry results for the 15 m device case, but considerably lower for the others. We note that if flow cytometry missed any of these cell types due to low EpCAM expression, it would only widen the disparity. Instead, we contend that the comprehensive manner in which scRNA-seq identifies cell types is superior to flow cytometry, particularly when a clear positive biomarker for all cell subpopulations is lacking. Flow cytometry is better suited to cell counting, however, and based on those results, device processing consistently produced comparable numbers of cells at 15 min and at least 50% more cells at 60 min, relative to the 60 min control. We used these estimates as weighting factors (1× for 15 min, 1.5× for 60 min), along with percentages in Fig. 4b, to calculate aggregate device platform yields (see Supplementary Information, Supplementary Table S3). Total endothelial cell recovery was ~4-fold greater than the control, while other cell types were ~2- to 2.5-fold greater, which all match flow cytometry (Fig. 3a–c). While the true weighing factors may be slightly different, it does appear that the relative numbers between control and device platform are consistent between flow cytometry and scRNA-seq. However, the relative numbers across cell types varies considerably, which may have resulted from biasing during FACS collection or droplet loading in the 10X Chromium system, which have been documented previously[62]. Our results suggest a preferential selection of endothelial cells and leukocytes during these

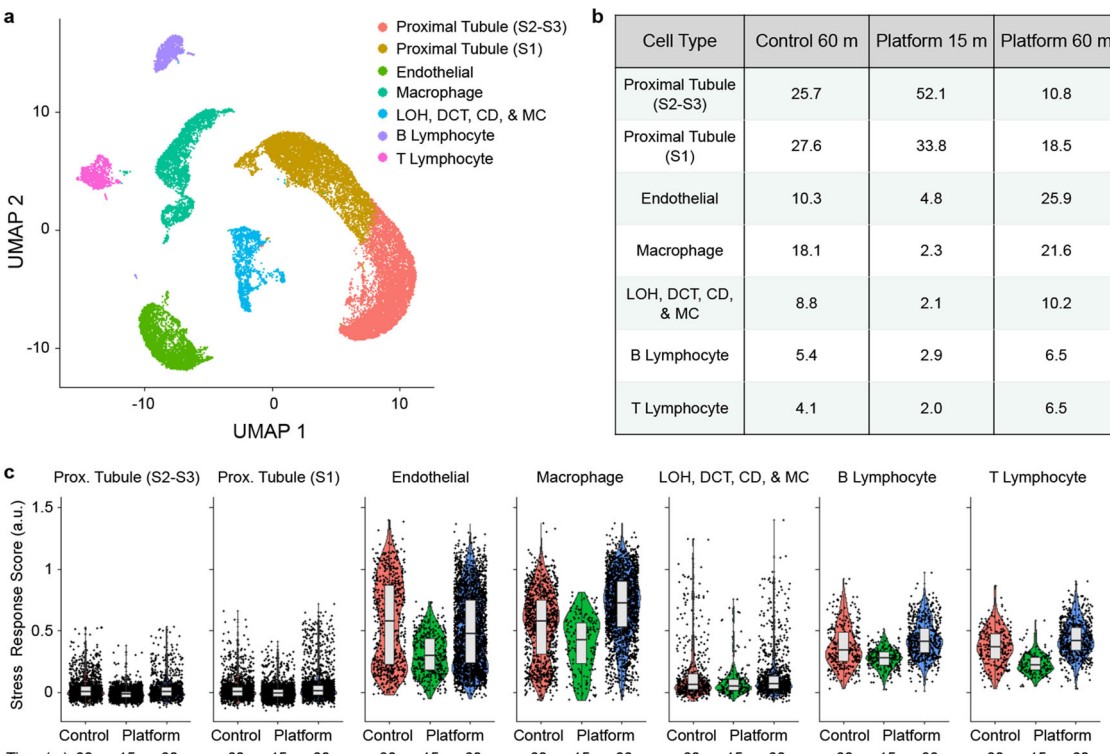

**Fig. 4 scRNA-seq of murine kidney ($n = 1$).** Cell suspensions obtained from the microfluidic platform at 15 and 60 min intervals, as well as the 60 min control, were sorted by FACS to remove dead cells and debris, loaded onto a 10X Chromium chip, and sequenced at >50,000 reads/cell. **a** UMAP displaying seven cell clusters that correspond to two different proximal tubule subtypes, endothelial cells, macrophages, B lymphocytes, and T lymphocytes. The seventh cluster contained a mixed population corresponding to distal convoluted tubules (DCT), loop of Henle (LOH), collecting duct (CD), and mesangial cells (MC). **b** Population distributions for each cell cluster and processing condition. Proximal tubules were predominantly eluted from the microfluidic platform in the 15 min interval, while endothelial cells and macrophages were enriched in the 60 min interval. **c** Stress response scores, displayed in violin plots, were generally lower for the 15 min device interval. Horizontal line depicts median and edges of box depict 1$^{st}$ and 3$^{rd}$ quartiles.

steps. Nevertheless, we conclude that our microfluidic device platform can address cell-specific biasing of kidney tissue during isolation by enriching endothelial cells, which have been shown to be underrepresented using standard tissue dissociation workflows[34], while maintaining all other cell subtypes at similar numbers. We note that only a few potential podocytes were observed in either control or device samples (see Supplementary Information, Supplementary Fig. S8), which may be attributed to the fact that we only utilized collagenase for enzymatic digestion. Kidney atlases prepared using Liberase also lacked podocytes[34], while the combination of collagenase and Pronase, as well as a cold-active protease, yielded podocyte cell clusters[36]. This indicates that the choice of enzyme is still important even in settings with enhanced mechanical forces.

Lastly, we evaluated stress response genes, which can interfere with cell identification using transcriptomic information. Induction of stress responses has been linked to conventional tissue dissociation protocols[34,36,38,39]. Since a large number of genes have been implicated, we calculated a stress response score based on a published set of 140 stress response genes (Fig. 4c) that were reported in previous scRNA-seq work[39,63]. We found that stress response scores were cell type specific, with proximal tubules exhibiting the lowest values, as recently observed[38]. Stress response scores were generally lower for the 15 min interval condition compared to the 60 min interval and control cases. This is consistent with previous findings that shortening enzymatic digestion time reduces dissociation-induced transcriptional artifacts[36,38]. Importantly, we found no evidence that exposure to fluid shear stresses within the digestion device heightened the

stress response for any cell type. This suggests that time was the predominant factor, which can be mitigated using the interval concept offered by our microfluidic platform. Expression values for selected stress response genes are individually shown in the Supplementary Information, Supplementary Fig. S9.

Solid tumors can exhibit high degrees of intratumoral heterogeneity, which has been directly implicated in cancer progression, metastasis, and the development of drug resistance[12,13]. This heterogeneity has successfully been captured using scRNA-seq and linked to survival for glioblastoma, drug resistance in melanoma, and prognosis for colorectal cancer[3,63–67]. Moreover, it is expected that expanded application of scRNA-seq in clinical settings will soon emerge to provide molecular and cellular information for guiding personalized therapies[68]. Due to abnormal extracellular matrix composition and density, however, tumor tissues are considered to be amongst the most difficult epithelial tissues to dissociate[55,69,70]. We evaluated microfluidic processing of mammary tumors that spontaneously arise in MMTV-PyMT transgenic mice. We first optimized the minced digestion and integrated dissociation/filter devices separately. The digestion device generated ~2-fold more EpCAM+ epithelial cells than the controls after 15 and 30 min, and the difference extended to 2.5-fold after 60 min (see Supplementary Information, Supplementary Fig. S10a). Viability was higher for device processed samples than controls at all time points (see Supplementary Information, Supplementary Fig. S10b). We then tested the integrated dissociation/filter device and again found that a single pass was optimal (see Supplementary Information, Supplementary Fig. S10c, d). In this case, recirculation for 1 and 4 min produced similar cell numbers, but with lower viability.

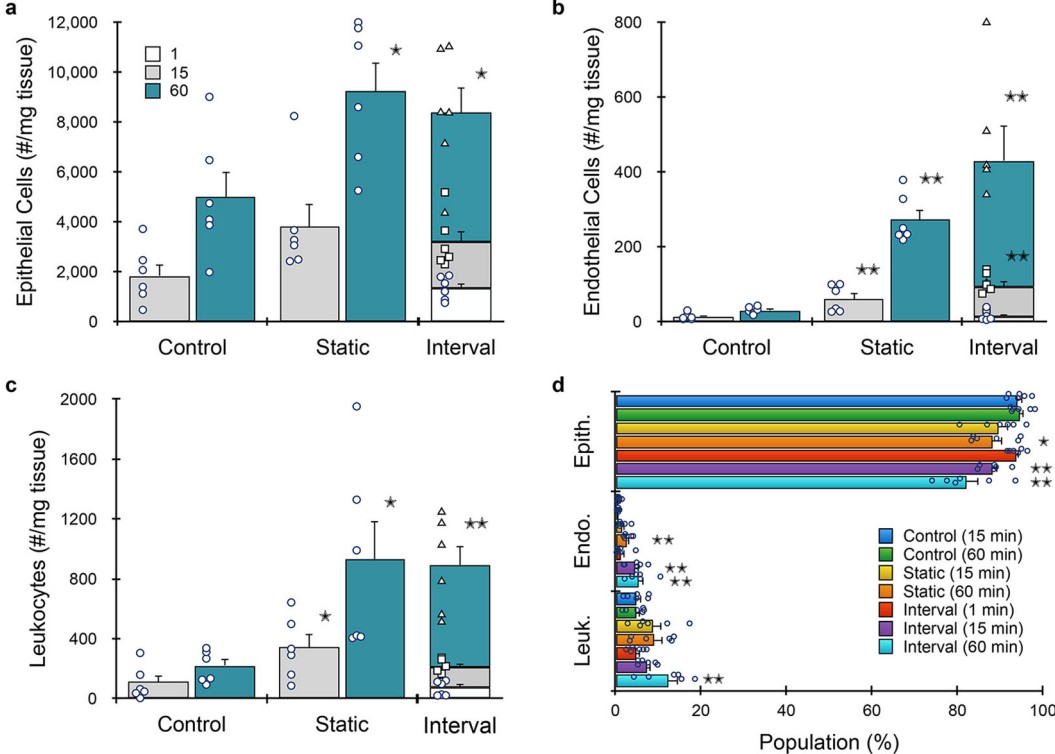

**Fig. 5 Microfluidic platform results for murine breast tumor.** Breast tumors from MMTV-PyMT mice ($n = 6$) were resected, minced, processed with the microfluidic platform, and analyzed by flow cytometry. **a** EpCAM+ epithelial cells were ~2-fold higher at both time points. **b** Endothelial cells were enhanced even more by the microfluidic platform, with five- and ten-fold more cells recovered after 15 and 60 min, respectively. **c** Leukocytes increased by three- and five-fold after 15 and 60 min, respectively. The interval format produced similar total cell numbers relative to the corresponding static time point, except for endothelial cells, which were slightly higher. **d** Population distributions obtained for each processing condition. Device processing enriched both endothelial cells and leukocytes at all but the 1 min time point. Data are presented as mean values ± SEM from at least three independent experiments. Circles indicate values for experimental replicates. For stacked plots, experimental replicates are indicated by circles at 15 min, squares at 30 min, and triangles at 60 min. Two-sided $T$ test was used for statistical testing. Stars indicate $p < 0.05$ and double stars indicate $p < 0.01$ relative to the control at the same digestion time. $p$ values for all comparisons are presented in the Source Data file.

Results for the full microfluidic device platform are shown in Fig. 5, and were generally similar to kidney, but with two- to three-fold lower cell counts/mg tissue overall. However, the device platform still produced significantly more cells than controls. Epithelial cells were ~2-fold higher at both time points (Fig. 5a). Endothelial cells were again liberated more effectively by device processing, with five-fold more cells recovered after 15 min and ten-fold more after 60 min (Fig. 5b). Leukocytes increased by three- and five-fold after 15 and 60 min, respectively (Fig. 5c). The interval format produced similar total epithelial cell and leukocyte numbers when compared to the corresponding static time point. However, ~30% more endothelial cells were obtained from intervals. We also note that a remarkably large number of epithelial cells (>15%) were obtained at the 1 min interval. Relative population percentages are shown in Fig. 5d. Device processing enriched for endothelial cells and leukocytes at all but the 1 min time point, which remained similar to controls. As with kidney, microfluidic processing was associated with higher batch-to-batch reproducibility, as measured by the coefficient of variation (see Supplementary Information, Supplementary Table S4). Viability for all three cell types were similar to the 15 min control and exceeded the 60 min control (see Supplementary Information, Supplementary Fig. S11). Thus, the microfluidic platform liberated more single cells from tumor, while also better preserving cell viability.

We then performed scRNA-seq, again using the 15 and 60 min device intervals and the 60 min control. A total of 24,527 cells were sequenced at an average of ~45,000 reads per cell. We identified six clusters corresponding to epithelial cells, macrophages, endothelial cells, T lymphocytes, fibroblasts, and granulocytes (Fig. 6a). Epithelial cells were the predominant cell population, representing 62.0% of control cells (Fig. 6b). Epithelial percentages increased slightly in the 15 min interval and decreased in the 60 min interval. We identified three sub-clusters within the epithelial population corresponding to luminal, basal, and proliferating luminal cells based on expression of *Krt14*, *Krt18*, and *Mki67*, respectively (see Supplementary Information, Supplementary Fig. S12). The luminal subtype dominated, as expected for MMTV-PyMT tumors. The basal subpopulation was enriched with device processing, while the proliferating luminal subpopulation was underrepresented. These results suggest that basal epithelium is more difficult to dissociate. Comparing cell populations between scRNA-seq and flow cytometry was more straightforward since EpCAM, CD45, and CD31 all correlated well with the expected cell types (see Supplementary Information, Supplementary Fig. S13). However, fibroblasts were not detected by flow cytometry, and account for a significant portion of the 60 min device condition. As with kidney, tumor epithelial percentages were significantly higher in flow cytometry data, which would further suggest biasing during sorting and/or droplet encapsulation. If we combine the population percentages in Fig. 6b with the same weighting factors used for kidney (1× for 15 min, 1.5× for 60 min), we can again calculate aggregate device platform yields (see Supplementary Information, Supplementary

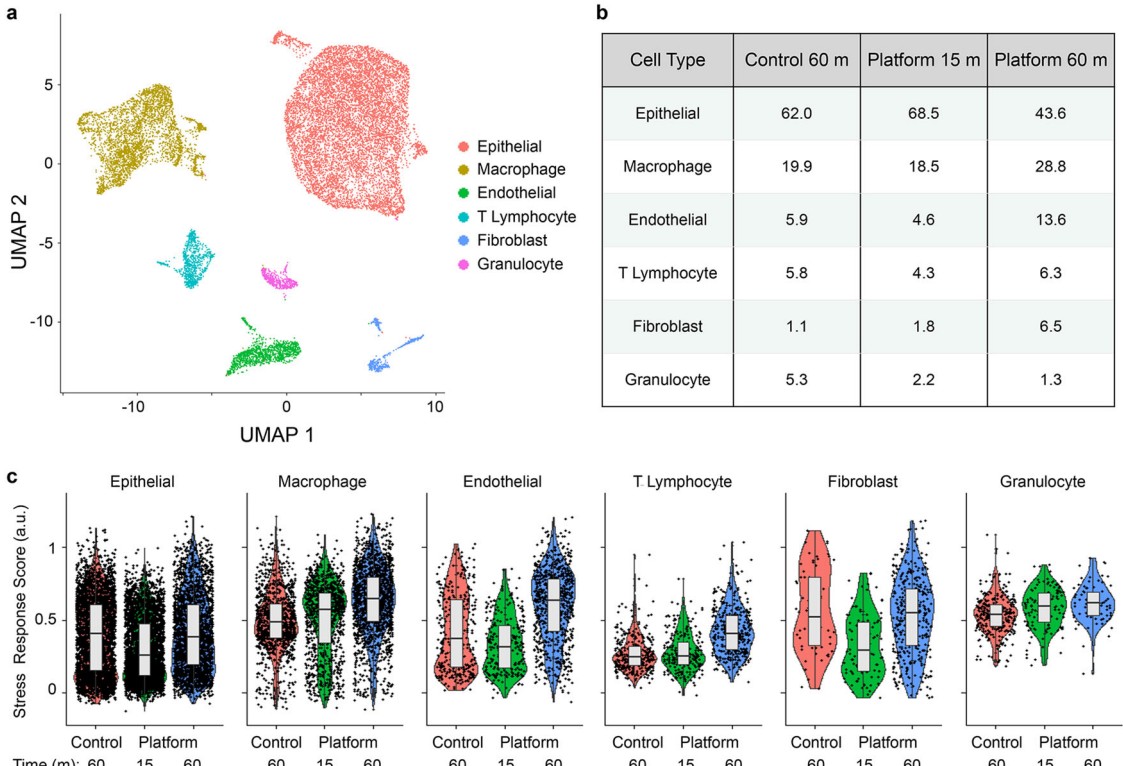

**Fig. 6 scRNA-seq of murine mammary tumor ($n = 1$).** Cell suspensions obtained from the microfluidic platform at 15 and 60 min intervals, as well as the 60 min control, were processed and analyzed using similar methods to kidney. **a** UMAP displaying six cell clusters that correspond to epithelial cells, macrophages, endothelial cells, T lymphocytes, fibroblasts, and granulocytes. **b** Population distributions for each cell cluster and processing condition. Epithelial cells were predominantly eluted from the microfluidic platform in the 15 min interval, while endothelial cells and fibroblasts were enriched in the 60 min interval. Fibroblasts were enriched in both platform conditions, while granulocytes were depleted. **c** Stress response scores, displayed in violin plots, were generally similar across conditions and cell types. Horizontal line depicts median and edges of box depict 1st and 3rd quartiles.

Table S5). Differences for the device aggregate relative to the control were ~2-fold for epithelial cells and 2.5- to 3-fold for T lymphocytes and macrophages, which are all similar to flow cytometry results (Fig. 5a, c). Endothelial cells were ~4-fold greater for the device platform, which is lower than the 10-fold difference from flow symmetry (Fig. 5b). Notably, fibroblasts were enriched by 10-fold using the device platform. Our results confirm that tissue processing with the microfluidic device platform can improve isolation of all cell types by at least 2.5-fold, as well as difficult to liberate cell types such as endothelial cells, fibroblasts, and basal epithelium by 4- to 10-fold.

Finally, we determined stress response scores as described for kidney. The importance of stress responses can be heightened for tumor since some response genes, such as members of the *Jun* and *Fos* families, have been associated with metastatic progression and drug resistance[38,71–73]. Stress response scores were similar across all cell types and conditions for tumor (Fig. 6c). It is possible that tumor cells are more sensitive to dissociation-induced transcriptional changes, and that even shorter intervals would be necessary to lower these responses. Expression values for selected stress response genes are individually shown in the Supplementary Information, Supplementary Fig. S14.

The liver plays a major role in drug metabolism and is frequently assessed for drug-induced toxicity. In fact, liver damage is one of the leading causes of post-approval drug withdrawal[22,74,75]. Thus, in vitro screening of drugs against primary liver tissue is a critical component of preclinical testing. Increasingly, organ-on-a-chip systems are being employed to better maintain hepatocyte functionality and activity in culture settings and to enable personalized testing on patient-derived primary cells[27,76]. While liver is softer and generally easier to dissociate, hepatocytes are well known to be fragile, and thus liver presents a unique dissociation challenge[77]. As such, we hypothesized that shorter device processing times would be effective for liver. For these experiments, murine liver was minced into ~1 mm³ pieces and hepatocytes were detected based on ASGPR1 expression. We first processed liver using the minced digestion device for either 15 or 60 min. After 15 min, hepatocyte recovery was ~4-fold higher for the device than the comparable control (Fig. 7a). Continued digestion of the control increased hepatocyte numbers further. Counterintuitively, continued processing in the digestion device diminished hepatocyte yield by approximately half. We believe this finding was caused by the combination of two factors: softer liver tissue is effectively broken down at earlier time points and fragile hepatocytes are more sensitive to damage from recirculation. We also tested a single pass through the integrated dissociation/filtration device, and found that hepatocyte recovery decreased. This was likely due to the large size of hepatocytes (~30 µm), which caused them to be retained or damaged by the 15 µm membrane. It also appears that damage may have been additive, as viability dropped to 45% after 60 min digestion device treatment and passing through the integrated device, while all other conditions were ~80% (Fig. 7b). Removing the 15 µm filter from the integrated dissociation/filter device increased hepatocytes by 30% relative to the digestion device alone, and by nearly three-fold relative to the control, while maintaining viability (see Supplementary Information, Supplementary Fig. S15).

Based on the initial optimization studies, we concluded that the microfluidic device platform should utilize short processing times

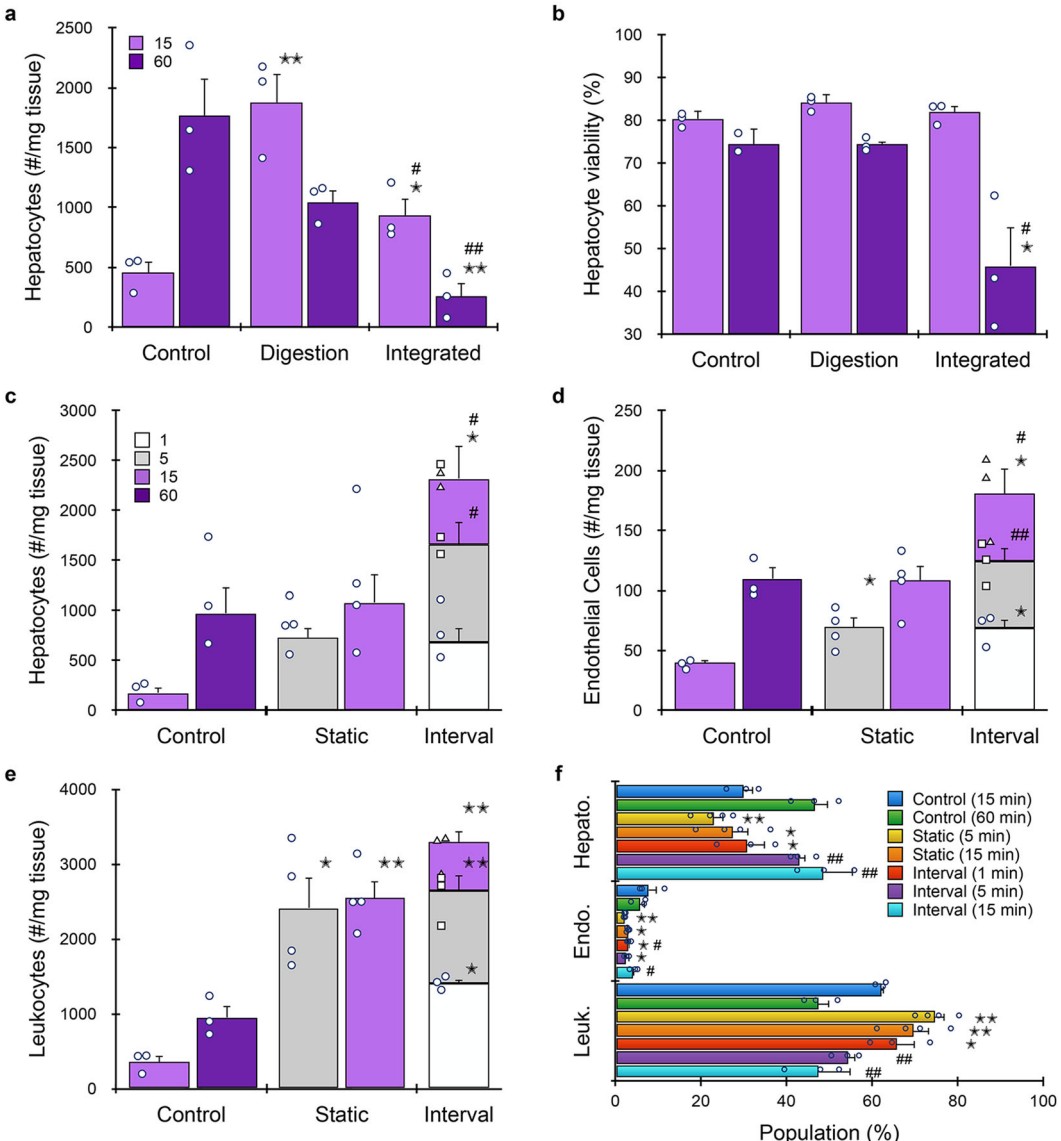

**Fig. 7 Microfluidic platform results for murine liver. a, b** Liver (*n* = 3 or 4 independent samples) was harvested, minced, and evaluated with the minced digestion device alone and in combination with the integrated dissociation/filter device. Hepatocytes were identified and quantified by flow cytometry. **a** The digestion device increased hepatocyte recovery by ~4-fold at 15 min, but continued digestion and passing through the integrated dissociation/filter device one time decreased hepatocyte yield, likely due to the large size and fragile nature of hepatocytes. **b** Hepatocyte viability was ~75–80% for all conditions, except the 60 min integrated condition. **c–f** Results using shorter digestion times and a single pass with a dissociation/filtration device containing only the 50 µm filter. **c** After only 5 min of microfluidic processing, four-fold more cells were obtained than the 15 min control and only slightly less than the 60 min control. Interval recovery enhanced hepatocyte yield by ~2.5-fold relative to the 60 min control and 15 min static conditions. The 1 min interval contributed substantially, producing ~70% as many hepatocytes as the 60 min control. Similar results were observed for **d** endothelial cells and **e** leukocytes, although the benefit of intervals was less pronounced. **f** Population distributions obtained for each processing condition. Microfluidic processing generally enriched for leukocytes, although there was a shift to hepatocytes for the later intervals. Data are presented as mean values ± SEM from at least three independent experiments. Circles indicate values for experimental replicates. For stacked plots, experimental replicates are indicated by circles at 15 min, squares at 30 min, and triangles at 60 min. Two-sided *T* test was used for statistical testing. Stars indicate *p* < 0.05 and double stars indicate *p* < 0.01 relative to the 60 min control. Cross-hatches indicate *p* < 0.05 and double cross-hatches indicate *p* < 0.01 relative to the static condition at the same digestion time. *p* values for all comparisons are presented in the Source Data file.

and the modified dissociation/filter device with only the 50 µm filter. After 5 min digestion device processing, ~700 hepatocytes were recovered/mg liver tissue (Fig. 7c). This was four-fold higher than the 15 min control and just slightly less than the 60 min control (~1000 hepatocytes/mg). Increasing digestion device processing time to 15 min enhanced hepatocyte recovery by 40%, to the same level as the 60 min control. The most striking results were observed under the interval format. After only 1 min, ~700 hepatocytes/mg tissue were recovered. Adding the 5 and 15 min

intervals resulted in ~2400 hepatocytes/mg, for a ~2.5-fold enhancement relative to both the 60 min control and 15 min static conditions. Hepatocyte viability remained at 90% for controls and most device conditions (see Supplementary Information, Supplementary Fig. S16a). Similar trends were observed for endothelial cells (Fig. 7d) and leukocytes (Fig. 7e), including significant recovery from the 1 min interval and enhanced overall cell numbers using the interval format. For endothelial cells, interval operation was ~1.5-fold higher than the 60 min control

and 15 min static device cases. For leukocytes, static device operation produced >2.5-fold more cells than the 60 min control, and interval operation further enhanced recovery to ~3.5-fold. Given the strong performance of the device platform with leukocytes and their relative abundance in liver compared to kidney and tumor, cell suspensions were enriched for leukocytes in comparison to the 60 min control (Fig. 7f). This was particularly true for the static time points and the 1 min interval. Interestingly, the three interval conditions contained very different representations of hepatocytes and leukocytes, suggesting that the choice of elution time could serve as a means to crudely select for one population over the other, if that was so desired. Batch-to-batch reproducibility was highest for microfluidic processing using intervals for all but endothelial cells (see Supplementary Information, Supplementary Table S6). Viability for endothelial cells and leukocytes remained similar to or greater than controls (see Supplementary Information, Supplementary Fig. S16b, c).

Taken together, the performance of the microfluidic processing platform with liver was quite unique relative to kidney and tumor. We believe that this caused by the fact that fluid shear forces are needed to break down tissue, but can also damage some cell types that have already been liberated. All tissues require proper balancing of these effects. For softer tissues like liver, the balance must be shifted away from breakdown and toward preservation, particularly for sensitive hepatocytes, which can be accomplished using interval recovery. Endothelial cells and leukocytes also exhibited some sensitivity to over-processing, although to a lesser degree. It is unclear whether this finding can be generalized to other tissues, including kidney and tumor. Liver sinusoidal endothelial cells are highly specialized, with abundant fenestrae and no underlying basement membrane[78]. These properties could also make sinusoidal endothelial cells particularly sensitive to damage. For leukocytes, we did not distinguish between those that originated within the liver, which would predominantly be Kupffer cells, from those that came from blood, which may be less sensitive to shear. Future studies directly assessing Kupffer cells, as well as hepatic stellate cells, would be of high interest, particularly to make progress toward complex liver models that utilize multiple cell types[74,79–81].

Heart failure is another leading cause of drug withdrawal from the market, combining with liver failure to account for ~70% of withdrawals[74,75]. Thus, there is robust interest in developing heart-on-chip technologies using primary cardiomyocytes for preclinical drug screening[30,74,75,82,83]. Cardiomyocytes have been shown to be highly sensitive to mechanical and enzymatic dissociation techniques[84,85]. In addition, they are disproportionately long in one direction, on the order of 100 μm or more[86]. For these experiments, murine heart was minced into ~1 mm³ pieces and cardiomyocytes were detected based on Troponin T expression. Since Troponin T is an intracellular marker, we used a fixable viability dye, Zombie Violet, in place of 7-AAD. Given potential concerns about cardiomyocyte size and shape, we first tested the effect of filter pore size in the integrated dissociation/filtration device. After 15 min processing with the minced digestion device, sample was passed through the original integrated dissociation/filter device with both 50 and 15 μm pore size membranes or the modified version with only the 50 μm membrane. Cell numbers and viability were similar for all conditions (see Supplementary Information, Supplementary Fig. S17), and thus we selected to use the original version with both membranes for heart tissue.

Next we evaluated the full microfluidic platform at different digestion times. We again focused on shorter processing times due to the potential sensitivity of cardiomyocytes. After 5 min treatment with the digestion device, ~2000 cardiomyocytes were recovered per mg heart tissue (Fig. 8a). This was lower than both the 15 and 60 min controls, by ~half and one-third, respectively.

Increasing digestion device processing to 15 min increased recovery to ~12,000 cells/mg, which was ~2-fold higher than the 60 min control. As with kidney, the interval format further increased cardiomyocyte recovery to ~18,000 cells/mg. Endothelial cell (Fig. 8b) and leukocyte (Fig. 8c) yields from the microfluidic device platform were significantly lower than the 60 min control. The interval format did improve recovery for both cases, but the 60 min control remained higher by ~2-fold for endothelial cells and ~1.5-fold for leukocytes. Based on this differential recovery, device platform processing resulted in significant enrichment of cardiomyocytes (Fig. 8d). Batch-to-batch reproducibility was highest for microfluidic processing using intervals (see Supplementary Information, Supplementary Table S7). Viabilities for all three cells types were similar to controls (see Supplementary Information, Supplementary Fig. S18). Considering results for all tissues in a comprehensive manner, heart likely lies in between the kidney/tumor and liver extremes. The tissue is still challenging to break down, which is why recovery was low at the early time points. Digestion was likely to be particularly ineffective on its own for cardiomyocytes due to strong intracellular connections formed by desmosomes and adherins junctions, while the microfluidic platform may provide the shear stresses necessary to break these connections and separate cardiomyocytes. However, the sensitivity of cardiomyocytes to mechanical damage is a confounding factor, which makes longer digestion times unlikely to improve results. Endothelial cells can arise from both vessels and the endocardium that lines the walls of the atrial and ventricular chambers. We contend that endocardium was liberated effectively by digestion alone since the chambers can be readily accessed by collagenase. As seen for kidney and tumor, however, blood vessels require longer time for effective release of endothelium, even with the device platform. This suggests that our results were likely dominated by endocardium, and that damage was the predominant reason for reduced recovery. The fact that interval recovery improved results for all cell types assessed in both heart and liver indicates that this mode is critical for optimal performance. In fact, temporal resolution should likely be increased, or ideally, be continuous, to prevent cell damage. Nevertheless, the microfluidic platform as currently configured and operated in this study consistently improved the recovery of single cells from diverse tissue types based on increased total cell yield, decreased processing time, and in some cases, both.

We have presented a microfluidic tissue processing platform comprised of a digestion device with features to facilitate loading and processing of minced specimens, as well as a dissociation/filter device that was integrated. Inclusion of all three device concepts completes the full tissue dissociation workflow so that single cell suspensions are immediately ready for downstream analysis or alternative use. The minced digestion device accelerated tissue breakdown and produced significantly more single cells than traditional methods, while the integrated dissociation/filter device increased yield further, while also increasing reproducibility and fully maintaining viability. A diverse array of tissue types was tested that exhibited a wide range of properties, and two different single-cell analysis methods, flow cytometry and scRNA-seq, were utilized. We also introduced a processing scheme, interval operation, which allowed us to extract single cells at different time points during microfluidic digestion. We found that tissues that were physically tougher and more robust, such as kidney and tumor, could be processed continuously, producing similar cell numbers in less time (15 vs 60 min) and ~2.5-fold more single cells in total. scRNA-seq further confirmed that endothelial cells, fibroblasts, and basal epithelial cells were all enriched by the microfluidic platform, with each increasing by 4- to 10-fold. In addition, we found that shorter digestion times were

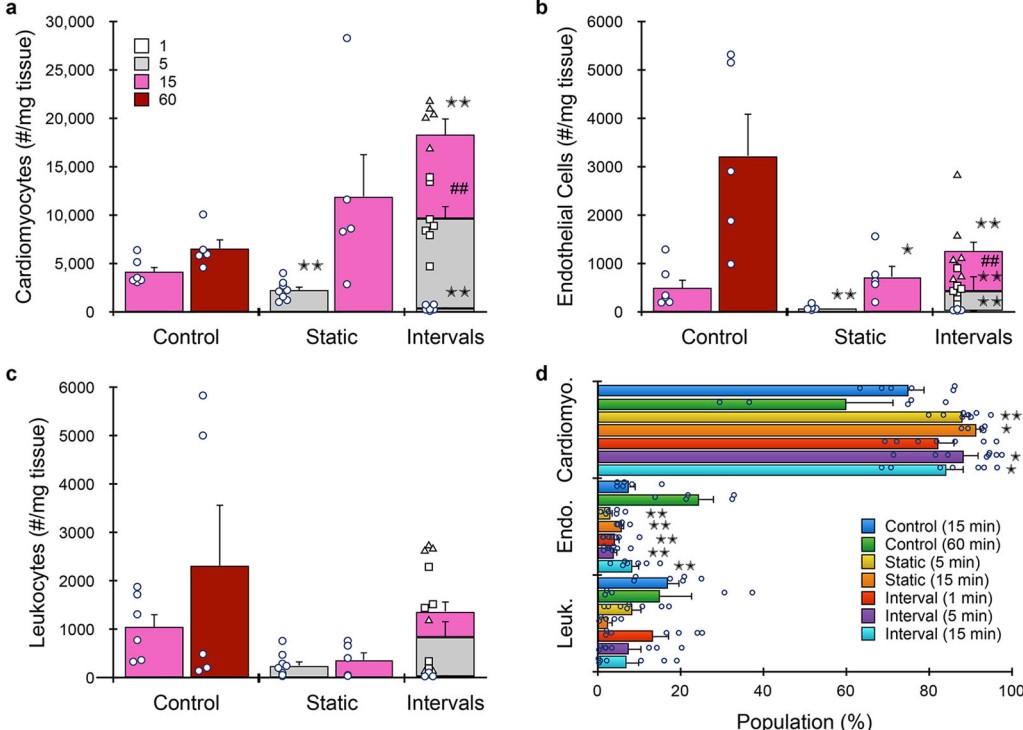

**Fig. 8 Microfluidic platform results for murine heart.** Hearts ($n = 5$ to 8 independent samples) were resected, minced, processed with the microfluidic platform (both 50 and 15 μm membranes), and analyzed by flow cytometry. We employed shorter digestion device time points due to the sensitivity of cardiomyocytes. **a** Microfluidic processing produced ~12,000 cardiomyocytes per mg after 15 min, which was ~2-fold higher than the 60 min control. Interval recovery produced optimal results again, increasing by ~50% and ~3-fold relative to the 15 min static and 60 min control conditions. **b** Endothelial cell and **c** leukocyte yields were significantly lower than the 60 min control under both static and interval formats. Interval recovery did improve results, but remained ~2-fold lower than the 60 min controls. **d** Population distributions obtained for each processing condition. Microfluidic processing generally enriched for cardiomyocytes. Data are presented as mean values ± SEM from at least three independent experiments. Circles indicate values for experimental replicates. For stacked plots, experimental replicates are indicated by circles at 15 min, squares at 30 min, and triangles at 60 min. Two-sided $T$ test was used for statistical testing. Stars indicate $p < 0.05$ and double stars indicate $p < 0.01$ relative to the 60 min control. Cross-hatches indicate $p < 0.05$ and double cross-hatches indicate $p < 0.01$ relative to the static condition at the same digestion time. $p$ values for all comparisons are presented in the Source Data file.

associated with lower stress responses for some cell types, but otherwise microfluidic processing did not add to the stress response in any case. These results clearly confirm that the microfluidic tissue processing platform holds exciting potential to advance scRNA-seq studies by reducing cell subtype-biasing, processing time, and/or stress response. For tissues that were softer and may contain sensitive cell types, like liver and heart, we found that processing times could be dramatically reduced and that interval operation was critical to avoid cell damage. These results will advance goals in tissue engineering and regenerative medicine, and could be particularly exciting for patient-derived organ-on-a-chip models for pharmacological studies. By focusing on minced specimens, our microfluidic tissue processing platform can readily be incorporated into the dissociation workflows for most, if not all, organs and tissues. Minimizing tissue pre-processing would be advantageous, and will be pursued in future work, along the lines of our original digestion device for larger tissue cores. Another future goal will be to further decrease interval recovery time points to better protect fragile cells, intentionally enrich certain cell subtypes, and minimize stress responses. Ideally, we would integrate a cell separation strategy that would make it possible to elute single cells from the platform as soon as they are generated. We will also evaluate more tissues, focusing on performance optimization across diverse properties and cell subtypes, as well as explore alternative proteolytic enzymes such as cold-active proteases[36,38]. An important

validation for all tissues will be to compare cell subtype numbers and ratios after dissociation to the ground truth determined using an alternate method, possibly comprehensive probe-based imaging. This will confirm recovery efficiency and biasing, and will be a major focus in future work. Finally, we envision incorporating microfluidic cell sorting and detection capabilities into the platform to create fully integrated and point-of-care technologies for cell-based diagnostics and drug testing, with a focus on human tissues for clinical applications.

## Methods

**Device fabrication**. Microfluidic minced digestion and integrated dissociation/filter devices were fabricated by ALine, Inc. (Rancho Dominguez, CA). Briefly, fluidic channels, vias, and openings for membranes, luer ports, and hose barbs were etched into PMMA or PET layers using a $CO_2$ laser. Nylon mesh membranes were purchased from Amazon Small Parts (15 and 50 μm pore sizes; Seattle, WA) as large sheets and were cut to size using the $CO_2$ laser. Device layers and other components (hose barbs, nylon mesh membranes) were then assembled, bonded using adhesive, and pressure laminated to form monolithic devices.

**Murine tissue models**. Kidney, liver, and heart were harvested from freshly sacrificed BALB/c or C57B/6 mice (Jackson Laboratory, Bar Harbor, ME) that were determined to be waste from a research study approved by the University of California, Irvine's Institutional Animal Care and Use Committee (courtesy of Dr. Angela G. Fleischman). Mammary tumors were harvested from freshly sacrificed MMTV-PyMT mice (Jackson Laboratory, Bar Harbor, ME). All mice were maintained at the University of California, Irvine according to the guidelines of the Institutional Animal Care and Use Committee. For kidneys, a scalpel was used to

prepare ~1 cm long x ~1 mm diameter strips of tissue, each containing histologically similar portions of the medulla and cortex. Tissue strips were then further minced with a scalpel to ~1 mm³ pieces. Liver, mammary tumor, and heart were uniformly minced with a scalpel to ~1 mm³ pieces. Minced tissue samples were then weighed and either processed with the devices as described below or processed as controls. Controls were placed within microcentrifuge tubes, digested at 37 °C in a shaking incubator under gentle agitation for 15, 30, or 60 min, and mechanically disaggregated by repeated pipetting and vortexing. 0.25% collagenase type I (Stemcell Technologies, Vancouver, BC) was used for both control and device processed conditions. Finally, cell suspensions were treated with 100 Units of DNase I (Roche, Indianapolis, IN) for 10 min at 37 °C and washed by centrifugation into PBS+1% BSA (PBS+).

**Minced digestion device operation.** A step-by-step protocol describing digestion device operation can be found at Protocols.io[87]. Minced digestion devices were prepared by affixing 0.05" ID tubing (Saint-Gobain, Malvern, PA) to the device inlet and outlet hose barbs, which was then connected to an Ismatec peristaltic pump (Cole-Parmer, Wertheim, Germany) with 2.62 mm ID tubing (Saint-Gobain, Malvern, PA). Prior to experiments, devices and tubing were incubated with SuperBlock (PBS) blocking buffer (Thermo Fisher Scientific, Waltham, MA) at room temperature for 15 min to reduce nonspecific binding of cells to channel walls and washed with PBS+. Minced pieces of tissue were loaded into the device tissue chamber through the luer inlet port. Devices and tubing were then primed with 0.25% collagenase type I solution (StemCell Technologies, Vancouver, BC), and the luer port was closed off using a stopcock. The experimental setup consisting of the device, tubing, and peristaltic pump was then placed inside a 37 °C incubator to maintain optimal enzymatic activity. The collagenase solution was recirculated through the device and tubing using the peristaltic pump at a flow rate of 10 or 20 mL/min for the specified time.

**Quantification of DNA recovered from cell suspensions.** Purified gDNA content of digested kidney tissue suspensions was assessed using a Nanodrop ND-1000 (Thermo Fisher, Waltham, MA) following isolation using a QIAamp DNA Mini Kit (Qiagen, Germantown, MD) according to the manufacturer's instructions. gDNA for device processed samples represents the cellular contents eluted from the device after operation, while gDNA for control samples represent the total amount of gDNA present in these samples.

**Cell aggregate studies.** MCF-7 human breast cancer cells were obtained from ATCC (Manassas, VA) and cultured as recommended. Prior to experiments, confluent monolayers were briefly digested for 5 min with trypsin-EDTA, which released cells with a substantial number of aggregates. Cell suspensions were prepared for experiments by centrifugation and resuspension in PBS+. MCF-7 cells were then recirculated through the peristaltic pump system alone, or through the system with a digestion or integrated dissociation/filter device attached using methods described in the main text. For the initial study, flow was recirculated only through the dissociation portion of the integrated device but not passed through the nylon filters of the filtration component for final sample collection in order to avoid confounding effects. To achieve this, the effluent outlet of the integrated device was closed off during pump operation using a stopcock. For all three tests, we used 5, 10, or 20 mL/min flow rates, and recirculation times of 0.5, 1, 4, and 10 min. Following experiments, devices and tubing were washed with 2 mL PBS+ to flush out and collect any remaining cells. Cell counts and viability were obtained both before and after recirculation using a Moxi Flow cytometer with type MF-S cassettes (Orfo, Hailey, ID) and propidium iodide staining.

**Integrated dissociation/filter device operation.** A step-by-step protocol describing device platform operation can be found at Protocols.io[87]. Following processing with the minced digestion device, tubing was connected from the outlet of the minced digestion device to the inlet of the integrated dissociation and filtration device. If recirculation was intended, tubing was connected from the cross-flow outlet to the peristaltic pump, while the outlet of the integrated device was closed off with a stopcock. Fluid was then pumped through the dissociation component at 10 mL/min flow rate. For final collection of the sample, or if only one pass through the dissociation component was utilized, the cross-flow outlet was closed off with a stopcock, and sample was pumped through at 10 mL/min and collected from the effluent outlet. Following all experiments, devices were washed with 2 mL PBS+ to flush out and collect any remaining cells. For time interval recovery, each PBS+ wash was followed by repriming of the device and tubing with collagenase solution, and the outlet of the minced digestion device was reconnected to the peristaltic pump for continued recirculation until the next collection period. Pictures of the fully integrated system, including pump, digestion, and dissociation/filter devices, are shown in the Supplementary Information, Supplementary Fig. S19 under recirculation and elution formats.

**Analysis of cell suspensions using flow cytometry.** A step-by-step protocol describing flow cytometry analysis for kidney samples can be found at Protocols.io[88]. Cell suspensions were analyzed using tissue specific flow cytometry panels shown in Table 1. For initial studies with kidney, cell suspensions were stained

concurrently with 5 μg/mL anti-mouse CD45-AF488 (clone 30-F11, BioLegend, San Diego, CA), 7 μg/mL EpCAM-PE (clone G8.8, BioLegend, San Diego, CA), and 5 μg/mL TER119-AF647 (clone TER-119, BioLegend, San Diego, CA) monoclonal antibodies for 30 min. Samples were then washed twice with PBS+ by centrifugation, stained with 3.33 μg/mL 7-AAD viability dye (BD Biosciences, San Jose, CA) on ice for at least 10 min, and analyzed on a Novocyte 3000 Flow Cytometer (ACEA Biosciences, San Diego, CA). Flow cytometry data was compensated using single stained cell samples or compensation beads (Invitrogen, Waltham, MA). Gates encompassing the positive and negative subpopulations within each compensation sample were used to calculate a compensation matrix in FlowJo (FlowJo, Ashland, OR). A sequential gating scheme (see Supplementary Information, Supplementary Fig. S20) was used to identify live and dead single epithelial cells, leukocytes, and red blood cells. Signal positivity was determined using appropriate fluorescence minus one (FMO) controls. Final studies with kidney, tumor, and liver used BV510 with CD45 (12.5 μg/mL, BioLegend, San Diego, CA) and also incorporated 8 μg/mL CD31-AF488 for endothelial cells. Liver demonstrations replaced EpCAM-PE with 10 μg/mL ASGPR1-PE (clone 8D7, Santa Cruz Biotechnology, Dallas, TX) for hepatocytes. Heart demonstrations used 1:1000 dilution of Zombie Violet (BioLegend, San Diego, CA) instead of 7-AAD for viability, and replaced EpCAM-PE with 0.15 μg/mL Troponin T-PE (clone REA400, Milentyi Biotec, San Diego, CA) for cardiomyocytes.

**Flow cytometry gating protocol.** Cell suspensions obtained from digested murine kidney, mammary tumor, liver, and heart samples were stained with the fluorescent probes listed in Table 1 and analyzed using flow cytometry. Acquired data were compensated and assessed using a sequential gating scheme (Supplementary Information, Supplementary Fig. S20). Gate 1 was based on FSC-A vs SSC-A, and was used to exclude debris near the origin. Gate 2 was used to select single cells based on FSC-A vs FSC-H. Gate 3 distinguished leukocytes based on CD45-BV510 positive signal and TER119-AF647 negative signal, while red blood cells were identified based on TER119-AF647 positive signal and CD45-BV510 negative signal. Gate 4 was applied to the CD45(−)/TER119(−) cell subset and used to identify epithelial cells in kidney and tumor samples based on positive EpCAM-PE signal, hepatocytes in liver samples based on positive ASGPR1-PE signal, and cardiomyocytes in heart samples based on positive Troponin T-PE signal. Gate 5 was applied to the EpCAM(−) cell subset in kidney and tumor samples, the ASGPR1(−) cell subset in liver, and the Troponin T(−) cell subset in heart tissue to identify endothelial cells based on positive CD31-AF488 signal. Finally, gate 6 was used to identify live cells in epithelial, hepatocyte, cardiomyocyte, leukocyte, and endothelial cell subsets based on negative 7-AAD or Zombie Violet (heart) signal. Appropriate isotype controls were initially used to assess nonspecific background staining, and appropriate FMO controls were used to determine positive signal cutoffs and set gates. Control samples were left unstained.

**Single-cell RNA sequencing.** A step-by-step protocol describing scRNA-seq can be found at Protocols.io[89]. These studies used mice (12 weeks, male, C57BL/6 for kidney; 10 weeks, female, MMTV-PyMT for mammary tumor, both from Jackson Laboratory, Bar Harbor, ME), which were euthanized by CO₂ inhalation. Kidneys and mammary tumors were dissected, minced into ~1 mm³ pieces, and prepared as described for the microfluidic platform (15 and 60 min digestion device intervals, single pass through integrated dissociation/filter device) or control (60 min digest) using 0.25% type I collagenase. Recovered cells were centrifuged (400×g, 5 min), treated with 100 Units of DNase I for 5 min at 37 °C, and washed by centrifugation into PBS+. Samples were then incubated with RBC lysis buffer for 5 min on ice, centrifuged, and resuspended in PBS+. Cells were stained with SytoxBlue (Life Technologies, Carlsbad, CA, USA) prior to FACS (FACSAria Fusion, BD Biosciences, Franklin Lakes, NJ) to remove dead cells and ambient RNA. Sorted live single cells (SytoxBlue-neg) were centrifuged and resuspended at a concentration of 1000 cells/μL in PBS with 0.04% BSA. The 10X Chromium system (10x Genomics, Pleasanton, CA) was then used for droplet-enabled scRNA-seq. Oil, cells, reagents, and beads were loaded onto an eight-channel microfluidic chip. Lanes were loaded with ~17,000 cells from each of the samples, determined using an automated cell counter (Countess II, Invitrogen, Carlsbad, CA). Library generation for 10x Genomics Single Cell Expression v3 chemistry was then performed according to the manufacturer's instructions. An Illumina NovaSeq 6000 platform (Illumina, San Diego, CA) was used to sequence the samples at a depth of ~60,000 reads/cell for kidney and ~45,000 reads/cell for mammary tumor. Sequencing fastq files were aligned using 10x Genomics Cell Ranger software (version 3.1.0) to an indexed mm10 reference genome. Cell Ranger Aggr was used to normalize the mapped reads for cells across the libraries for each data set. Genes that were not detected in at least three cells were discarded from further analysis. Cells with low (<200) or high (>3000 for kidney; >4000 for mammary tumor) unique genes expressed were also discarded, as these potentially represent low quality or doublet cells, respectively[4]. Cells with high mitochondrial gene percentages were also discarded (>50% for kidney and >25% for mammary tumor), as these can also represent low quality or dying cells[90]. The Seurat pipeline was used for cluster identification[60], principle component analysis was performed using genes that are highly variable, density clustering was performed to identify groups, and Uniform Manifold Approximation and Projection plots were used to visualize the groupings. For kidney, cell clusters were annotated using two approaches. First, top differential

genes in each cluster were examined to determine the cell type of the cluster based on expression of known marker genes (e.g., *Kap*, *Napsa*, and *Slc27a2* for S2–S3 proximal tubules[36], *Gpx3* for S1 proximal tubules[36], *Emcn* for endothelial cells[34], *Slc12a1* for LOH[4], *Slc12a3* for DCT[4], etc.). Second, since well-established atlases of murine kidney were available, we used a cell scoring method[63] to compare marker gene signatures from each of our clusters to published datasets[4,91] to confirm cluster annotations (see Supplementary Information, Supplementary Fig. S5). For tumor, cell clusters were annotated by examining top differential genes in each cluster to determine cell type based on expression of known marker genes (e.g., *EpCAM* for epithelial cells). Cellular stress responses were assessed using the same previously developed scoring method to compare stress response gene expression from each cluster to a previously published data set of known stress response genes[39,63].

**Statistics**. Data are represented as the mean ± standard error. Error bars represent the standard error from at least three independent experiments. *P* values were calculated from at least three independent experiments using Student's *T* test. Coefficient of variation was calculated as the standard error divided by the mean to represent batch-to-batch reproducibility between experimental replicates.

**Reporting summary**. Further information on research design is available in the Nature Research Reporting Summary linked to this article.

## Data availability

The authors declare that all data supporting the findings of this study are available within the article and its Supplementary Information files. All RNAseq data matrices along with their associated meta data have been deposited in the GEO database under accession code GSE163508 and SRA database under accession code PRJNA685210. Source Data are provided with this paper.

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

## Acknowledgements

The authors would like to thank Dr. Angela G. Fleischman for kindly donating mouse kidney, liver, and heart tissues. We would like to thank Dr. Brianna Hoover and Hamad Alshetaiwi for preparing tissue samples. We would also like to thank Danny Duong, Dalia Hammouri, and William Juan for assisting with experiments and/or image analysis. Funding support was received from the National Science Foundation and the industrial members of the Center for Advanced Design and Manufacturing of Integrated Micro-fluidics (NSF I/UCRC award number IIP-1841509), the Chan Zuckerberg Initiative, an advised fund of Silicon Valley Community Foundation (award number 173894), as part of the Human Cell Atlas project, and the National Cancer Institute of the National Institutes of Health (award number P30CA062203). The content is solely the responsibility of the authors and does not necessarily represent the official views of the National Science Foundation or National Institutes of Health.

## Author contributions

J. A. L. and J. B. H. devised the concept for the work and designed the microfluidic device technologies. J. A. L., M. A., and Q. H. N. carried out the experimental work. All authors carried out the experimental analysis. J. A. L. and J. B. H. wrote the manuscript, which all authors reviewed and edited.

## Competing interests

J. B. H. is a co-founder of Kino Discovery, which is in the process of licensing intellectual property for the tissue processing devices. The other authors declare no competing interests.
