## [Peer Review File · Nature Communications]

Reviewers' Comments:

Reviewer #1:

Remarks to the Author:

The authors have developed microfluidic technology for generating single-cell suspensions from complex tissue specimens. They have demonstrated that the technology performs comparably to conventional dissociation methods across a wide variety of tissue contexts. Importantly, the authors purposefully chose tissue contexts that are notoriously challenging, which is laudable. They also demonstrated the compatibility and performance of the resulting suspensions with multiple single-cell analysis tools, including flow cytometry and single-cell RNA-seq (scRNA-seq). This is an interesting and timely approach that could potentially enhance the scalability and reproducibility of single-cell analysis tools that require tissue dissociation. I am generally enthusiastic about this manuscript, but I have some technical concerns that should be considered:

1) Throughout the manuscript, the main comparisons used to assess the performance of microfluidic dissociation are different microfluidic conditions (e.g. times) and controls with conventional dissociation. Comparisons between these methods were performed using both flow cytometry and scRNA-seq, and detailed descriptions and explanations for similarities and differences are provided. These comparisons are valuable because they place the performance and results from microfluidic dissociation in the context of more conventional methods. However, this part of the study lacks a "ground truth". All of the methods used in these comparisons require dissociation, and so we really don't know which methods perform better in terms of preserving the true cellular composition of the tissue. The best (albeit imperfect) way to assess this is imaging intact tissue, either with immunohistochemical analysis of protein markers or in situ hybridization of RNA markers. When we observe differences in cellular composition between microfluidic and conventional dissociation, it's difficult to assess which of them is giving more accurate information without comparison to intact tissue. Quantifying the fractional composition of a handful of key cell types with established markers by imaging tissue sections would improve this component of the study significantly.

2) The authors provide comparisons of conventional and microfluidic dissociation by scRNA-seq. These data are very useful. However, the comparisons provided are restricted mainly to compositional and gene expression analysis. It would be highly informative to systematically compare these methods using more basic performance metrics for scRNA-seq such as coverage (number of unique transcripts/UMIs detected per cell, number of genes detected per cell), alignment rate, multiplet rate (particularly crucial for comparing dissociation methods), and some measure of ambient RNA contamination (e.g. relative transcript counts between cell-associated barcodes and background-associated barcodes).

3) One potential promise of microfluidic approaches is reproducibility. In many of the experimental comparisons shown here, the authors include independent replicates. While these replicates are used to assess the statistical significance of differences in cellular composition between methods, the authors do not include a rigorous assessment of reproducibility. For example, it would be interesting to know if the microfluidic dissociations are more reproducible (or comparably reproducible) in terms of cellular composition across independent replicates compared to conventional dissociation.

Reviewer #2:

Remarks to the Author:

This paper is a continuation study by the authors to develop a microfluidic chip to process tissue samples into single cells. There is a broad interest in manipulating tissue slices to obtain single cells for a range of applications, and this is well covered in the manuscript. The paper is also well executed from an engineering point of view. The experiments are carefully reported and the

results seem robust. However, it is not clear what the novelty of the work over the author's earlier publications (e.g., refs 52, 53) beyond some minor points such as interval processing, integration, scRNA-seq. In the past the authors showed the utility of 2 chips individually to first treat the tissue with digestive enzymes (minced digestion device) and then use another chip to dissociate (integrated dissociation and filter device). In this paper, the claim is that they integrated the 2 chips. This is a simple engineering task. Moreover, both in the main figures and the supplementary material, there is no image of the integrated device to know how the 2 chips were integrated. The chip description states that they made 2 monolithic devices and then one assumes they are connected with a tube? The paper can benefit from a better description of the actual device and the advances over the existing platform by the authors. As it stands, the novelty of this paper is low. One new aspect is the analysis of the treated cells with scRNA-seq. However, this portion of the paper is not shedding any new light to some biological process that was not feasible to address in the past? They use scRNA information to further characterize the cell types upon digestion and dissociation. As much as the paper keeps using the words "novel" and "new", the fundamental concept is already extensively published by the authors and this paper reads as an incremental but solid engineering study that might be better targeted to an engineering journal.

Reviewer #3:

Remarks to the Author:

The manuscript by Lombardo et al describes a novel microfluidic platform which simplifies sample preparation for single-cell sequencing (and other purposes), as well as provides an improvement in cell recovery in comparison to the standard protocol. This is an important area of research and development, as existing approaches are labour-intensive and have numerous specific biases.

The study is well-designed, the manuscript is well-written, and the results mostly support authors' claims.

Concerns/suggestions:

1. The authors used stress response score to characterise the level of stress in dissociated cells. This is a valid approach driven by some of the known biases of tissue dissociation. However, I believe the manuscript would benefit if the authors could extend the comparison of gene expression profiles between the control and platform-obtained cells. Differential gene expression analysis performed on a cell type level would help to reveal potential artefacts (if any) beyond the handful of genes used as stress signature.

2. Would be useful to know how the platform use influences ambient RNA contamination levels. I.e. if it is increased/decreased when compared to the standard protocol - this can be seen e.g. in DE analysis suggested above.

3. I noticed that in kidney the authors did not label podocytes. It would be interesting to know if this cell population is present, but wasn't separated by the clustering, or whether it was lost entirely. Podocytes are potentially hard to recover because of their unusual shape. In Wu et al. atlas, podocytes were practically lost by the single-cell protocol, while in Adam et al. they were recovered by both CAP and 37C protocols. If podocytes are lost by the platform described in the manuscript, it would be important to state this and explain possible reasons.

4. I did not see any statement regarding whether raw/processed data have been made available to the community.

Reviewer #1

The authors have developed microfluidic technology for generating single-cell suspensions from complex tissue specimens. They have demonstrated that the technology performs comparably to conventional dissociation methods across a wide variety of tissue contexts. Importantly, the authors purposefully chose tissue contexts that are notoriously challenging, which is laudable. They also demonstrated the compatibility and performance of the resulting suspensions with multiple single-cell analysis tools, including flow cytometry and single-cell RNA-seq (scRNA-seq). This is an interesting and timely approach that could potentially enhance the scalability and reproducibility of single-cell analysis tools that require tissue dissociation. I am generally enthusiastic about this manuscript, but I have some technical concerns that should be considered:

1) Throughout the manuscript, the main comparisons used to assess the performance of microfluidic dissociation are different microfluidic conditions (e.g. times) and controls with conventional dissociation. Comparisons between these methods were performed using both flow cytometry and scRNA-seq, and detailed descriptions and explanations for similarities and differences are provided. These comparisons are valuable because they place the performance and results from microfluidic dissociation in the context of more conventional methods. However, this part of the study lacks a "ground truth". All of the methods used in these comparisons require dissociation, and so we really don't know which methods perform better in terms of preserving the true cellular composition of the tissue. The best (albeit imperfect) way to assess this is imaging intact tissue, either with immunohistochemical analysis of protein markers or in situ hybridization of RNA markers. If we observe differences in cellular composition between microfluidic and conventional dissociation, it's difficult to assess which of them is giving more accurate information without comparison to intact tissue. Quantifying the fractional composition of a handful of key cell types with established markers by imaging tissue sections would improve this component of the study significantly.

This is a very important point, and we fully acknowledge that knowing the ground truth of how many cells were initially present in the tissues before dissociation would be tremendously valuable, even if this information was only available for some cell subtypes. The ground truth would allow us to confirm whether any biasing has occurred and, most importantly, determine how close we are to recovering all cells. Because we lack this insight, we really only know how much better we are doing relative to conventional methods. While we believe that we have made a significant step toward this type of goal in this study, we also know that we are not there yet. Some aggregates are still eluted from our system even after passing through the three devices. There is also the potential that some cells are being lost to damage, for all tissue types not just liver and heart. We have balanced these effects here with these 3 devices to the best of

our ability, and will need new device concepts to minimize, or even eliminate, both aggregates and cell damage. We are working on such device concepts now.

It is important to note that obtaining a separate ground truth assessment will be very challenging. We agree that imaging is likely to be the best path, but entire organs would need to be imaged, piece by piece. Tissue sections are thin, but they do not represent a single layer cells, which will complicate counting. Numerous questions would need to be addressed regarding selection of protein vs RNA targets, to validate either flow cytometry and/or scRNA-seq. This would be a research undertaking unto its own. There are two primary branches to the Human Cell Atlas initiative, spatial methods (largely imaging) and single cell methods (largely scRNA-seq). It would be very exciting to merge them in our future work to help address this question of ground truth.

Although we cannot address the issue at this time, we believe it is important to convey this to our readers. Therefore, we have added the following passage to the Conclusion:

“An important validation for all tissues will be to compare cell subtype numbers and ratios after dissociation to the ground truth determined using an alternate method, possibly comprehensive probe-based imaging. This will confirm recovery efficiency and biasing, and will be a major focus in future work.”

2) The authors provide comparisons of conventional and microfluidic dissociation by scRNA-seq. These data are very useful. However, the comparisons provided are restricted mainly to compositional and gene expression analysis. It would be highly informative to systematically compare these methods using more basic performance metrics for scRNA-seq such as coverage (number of unique transcripts/UMIs detected per cell, number of genes detected per cell), alignment rate, multiplet rate (particularly crucial for comparing dissociation methods), and some measure of ambient RNA contamination (e.g. relative transcript counts between cell-associated barcodes and background-associated barcodes).

We agree that the referenced scRNA-seq metrics are important for comparison purposes, both within this work and to others, and have added it to Table S1. Coverage information relating to number of unique transcripts/UMIs detected per cell can be found in the “Median UMI” column. Coverage information (number of genes detected per cell) can be found in the “Median Gene” and “Total Gene” columns. Alignment rate information can be found in the “Reads Mapped Confidently to Genome”, “Reads Mapped Confidently to Exonic Regions”, and “Reads Mapped Confidently to Transcriptome” columns.

As for multiplet rate and ambient RNA contamination, we note that we stringently sorted single cells from aggregates and ambient RNA by FACS prior to running scRNA-seq. Thus, any metrics related to these factors would likely reflect the FACS process as much, or more, than the dissociation process. That said, we did add "Fraction of Reads in Cell" into Table S1, which provides information on ambient RNA contamination, with higher fractions suggesting less contamination. Values were generally high for all conditions.

3) One potential promise of microfluidic approaches is reproducibility. In many of the experimental comparisons shown here, the authors include independent replicates. While these replicates are used to assess the statistical significance of differences in cellular composition between methods, the authors do not include a rigorous assessment of reproducibility. For example, it would be interesting to know if the microfluidic dissociations are more reproducible (or comparably reproducible) in terms of cellular composition across independent replicates compared to conventional dissociation.

We wholeheartedly agree with this point. Alas, this can be a challenging concept to illustrate in a rigorous manner. For example, the work in this study was performed by a single highly trained and experienced researcher. We have observed anecdotally that new trainees obtain far more variable results with the controls than devices, but attempts to illustrate such points in a publication will come off as manufactured. Despite that this is an important issue, we have chosen to leave this point to future work.

Reviewer #2

This paper is a continuation study by the authors to develop a microfluidic chip to process tissue samples into single cells. There is a broad interest in manipulating tissue slices to obtain single cells for a range of applications, and this is well covered in the manuscript. The paper is also well executed from an engineering point of view. The experiments are carefully reported and the results seem robust. However, it is not clear what the novelty of the work over the author's earlier publications (e.g., refs 52, 53) beyond some minor points such as interval processing, integration, scRNA-seq. In the past the authors showed the utility of 2 chips individually to first treat the tissue with digestive enzymes (minced digestion device) and then use another chip to dissociate (integrated dissociation and filter device). In this paper, the claim is that they integrated the 2 chips. This is a simple engineering task. Moreover, both in the main figures and the supplementary material, there is no image of the integrated device to know how the 2 chips were integrated. The chip description states that they made 2 monolithic devices and then one assumes they are connected with a tube? The paper can benefit from a better description of the actual device and the advances over the existing platform by the authors. As it stands, the novelty of this paper is low. One new aspect is the analysis of

the treated cells with scRNA-seq. However, this portion of the paper is not shedding any new light to some biological process that was not feasible to address in the past? They use scRNA information to further characterize the cell types upon digestion and dissociation. As much as the paper keeps using the words "novel" and "new", the fundamental concept is already extensively published by the authors and this paper reads as an incremental but solid engineering study that might be better targeted to an engineering journal.

We would like to start by explaining that the main goal of this work is to directly connect with end users, primarily in the single cell RNA sequencing and organ-on-a-chip communities, and not necessary the microfluidics field. Numerous design changes were made to the device technologies to improve performance and facilitate use by researchers that do not have microfluidics experience. Most importantly, we confirm that cell suspensions produced by our devices contain accurate representations of main cell subtypes. This confirmation was the main intention of the scRNA-seq study, and in fact we demonstrated that our devices liberated more of the most challenging cell subtypes, like endothelial cells and fibroblasts. We did not reveal new biology here, although we do plan to explore such findings in future work.

As for our devices, we would like to clarify a few points. We acknowledge that all 3 device concepts were previously published, but they had never been combined in any way, shape, or form in previous work. Thus, utilizing all 3 devices together results in a novel platform. Most importantly, this platform can perform all digestion, dissociation, and filtering steps of tissue processing steps on-chip, which is also entirely novel in the field. For this reason, we have now replaced the term "novel" with "integrated" in the title.

As for the device designs, each contains new elements that substantially improved performance and/or ease of use for less technically-savvy end users. The digestion device was completely redesigned, moving from a clunky device that we made in the lab to a stream-lined version that can easily be loaded with specimens and that was made by a commercial fabrication process. This has made the device easy to use by non-technical basic scientists and researchers. Additionally, prior to developing the minced digestion device, the original digestion device concept was tested with minced tissue specimens. This original design, however, when tested with minced tissue specimens led to catastrophic device failure from clogging by these smaller pieces of tissue. Thus, the minced digestion device had to be designed to address the unique challenges of working with smaller tissue specimens. The two downstream devices, for dissociation and filtering, were combined into a single device, which we acknowledge was a straightforward engineering task. But part of the reason this was straightforward was that we had the forethought to design them in a similar manner in the first place.

Nevertheless, we demonstrate how these devices can best be used together, which has considerable value.

As for how the devices were connected, we simply used tubing to between the digestion step and downstream dissociation/filtering, as stated in the Methods under the section titled “Integrated Dissociation/Filter Device Operation.”

We hope that this discussion helps to place the work in better context with respect to the new device elements, novel capabilities as a platform, and direct impact to non-technical researchers interested in studying tissue biology at the single cell level.

Reviewer #3

The manuscript by Lombardo et al describes a novel microfluidic platform which simplifies sample preparation for single-cell sequencing (and other purposes), as well as provides an improvement in cell recovery in comparison to the standard protocol. This is an important area of research and development, as existing approaches are labour-intensive and have numerous specific biases.

The study is well-designed, the manuscript is well-written, and the results mostly support authors' claims.

Concerns/suggestions:

1. The authors used stress response score to characterise the level of stress in dissociated cells. This is a valid approach driven by some of the known biases of tissue dissociation. However, I believe the manuscript would benefit if the authors could extend the comparison of gene expression profiles between the control and platform-obtained cells. Differential gene expression analysis performed on a cell type level would help to reveal potential artefacts (if any) beyond the handful of genes used as stress signature.

We would like to start by clarifying that the stress response score is based on a published set of 140 stress response genes, which may or may not be classified as a handful depending on whom one is speaking with. To provide a little more clarity, however, we have added expression results from 12 classic stress response genes (Jun and Fos families, Hsps, and others) to the SI for both kidney and breast tumor, broken by down cell type. We have edited the main text as follows:

“Since a large number of genes have been implicated, we calculated a stress response score based on a published set of 140 stress response genes (Fig. 4c) that were reported in previous

scRNA-seq work.^{39,63} We found that ... our microfluidic platform. Expression values for selected stress response genes are individually shown in the Supporting Information, Fig. S9.”

Additionally, while we agree that investigation of other genes could be interesting to identify possible artifacts, there is no class of genes that immediately lends itself to testing this hypothesis, and thus the investigation could quickly scale to a level that is difficult to manage. We do note that gene expression artifacts did not interfere with identification of cell clusters, which is the intended scope of this study.

2. Would be useful to know how the platform use influences ambient RNA contamination levels. I.e. if it is increased/decreased when compared to the standard protocol - this can be seen e.g. in DE analysis suggested above.

This is an interesting and important question, however, as stated in our response to Rev 1 on this topic, most ambient RNA would have been removed by FACS prior to running scRNA-seq. The “Fraction of Reads in Cell” is now provided in Table S1 to offer some insight into remaining ambient RNA contamination. We feel that in depth analysis of this issue will offer limited value since it will be confounded by FACS. We have also edited the main text as follows:

“scRNA-seq quality metrics are shown in Supplementary Information, Table S1, and were comparable across the conditions.”

3. I noticed that in kidney the authors did not label podocytes. It would be interesting to know if this cell population is present, but wasn't separated by the clustering, or whether it was lost entirely. Podocytes are potentially hard to recover because of their unusual shape. In Wu et al. atlas, podocytes were practically lost by the single-cell protocol, while in Adam et al. they were recovered by both CAP and 37C protocols. If podocytes are lost by the platform described in the manuscript, it would be important to state this and explain possible reasons.

This is an important point. We did search for podocytes in our data, but came up almost entirely empty for both control and device conditions. We have now added a new SI figure to clearly show this. We believe this result was due to our choice of enzyme, collagenase only, and not the devices, as podocytes were generally missing from both control and device conditions. As noted, previous work found podocytes were absent when Liberase was used, but did appear for collagenase+Pronase and cold active protease. We will investigate this further in future work. For now, we believe that even under the enhanced shear forces generated in our devices, the choice of enzyme is still of high importance for releasing cells. We have edited the main text as follows:

“We note that only a few potential podocytes were observed in either control or device samples (see Supporting Information, Fig. S8), which may be attributed to the fact that we only utilized collagenase for enzymatic digestion. Kidney atlases prepared using Liberase also lacked podocytes,³⁴ while the combination of collagenase and Pronase, as well as a cold active protease, yielded podocyte cell clusters.³⁶ This indicates that the choice of enzyme is still important even in settings with enhanced mechanical forces.”

4. I did not see any statement regarding whether raw/processed data have been made available to the community.

Single cell RNA-seq data have been deposited and made available via the Gene Expression Omnibus (GEO) and the Sequence Read Archive (SRA). The submissions are being processed now, and we will report our accession codes when they become available. We have added the following to the text:

“Data availability

The authors declare that all data supporting the findings of this study are available within the article and its supplementary information files or from the corresponding author upon reasonable request. All RNAseq data quantified data matrices along with their associated meta data have been deposited in the GEO database under accession code (GSE163508) and SRA database under accession code (PRJNA685210).”

Reviewers' Comments:

Reviewer #1:

Remarks to the Author:

The authors have done a good job of responding to my second comment. They have stated that a response to my first comment is beyond the scope of the study. While this might be reasonable for my first comment about establishing a ground truth, especially given the complex tissues chosen for study here, I found the response to my third comment about reproducibility surprising. The authors compare conventional and microfluidic dissociation across multiple metrics, but claim that an assessment of reproducibility, which is not analyzed in the paper, "will come off as manufactured". In my opinion, researchers who use scRNA-seq for large-scale profiling efforts in complex tissues will value reproducibility above almost any other performance metric.

Reviewer #2:

Remarks to the Author:

The authors' response to the major criticism that this is simply an incremental advance over the published series of papers by the same authors is not satisfactory. The authors claim a lot of "soft" operational optimization advances but nothing fundamentally important to report to a scientific community. The technology aspect, albeit it might have required some tweaking to integrate the 3 components, is still identical to the published work. Moreover, the integration was not create a single chip to truly integrate all functions but rather it was done by using standard tubes and lines to connect 3 devices together. The biological aspect is standard single cell sequencing and no new information is generated. It is also not clear what it means the main purpose of the paper is to contact with end-users. This paper might be more appropriate for a trade journal or a company advertisement but not a scientific paper.

Reviewer #3:

Remarks to the Author:

My concerns have been addressed in the revised version.

Reviewer #1

The authors have done a good job of responding to my second comment. They have stated that a response to my first comment is beyond the scope of the study. While this might be reasonable for my first comment about establishing a ground truth, especially given the complex tissues chosen for study here, I found the response to my third comment about reproducibility surprising. The authors compare conventional and microfluidic dissociation across multiple metrics, but claim that an assessment of reproducibility, which is not analyzed in the paper, "will come off as manufactured". In my opinion, researchers who use scRNA-seq for large-scale profiling efforts in complex tissues will value reproducibility above almost any other performance metric.

We apologize that confusion on our part led to an inadequate response about reproducibility. We focused entirely on inter-operator reproducibility, which we do believe will be reduced by our fluidic platform. However, this would need to be tested on different researchers, ideally with varying experience levels, and we were concerned that this would not be convincing (hence seem manufactured) unless an extremely large number of tests could be conducted. While this would lead to interesting results, it would be a significant research undertaking unto itself that will be explored in future work.

It is now clear to us that we can and should assess reproducibility amongst the experimental replicates. The standard error is indicative of reproducibility, but it is preferable to scale by the mean, which yields the coefficient of variation. We have calculated the COV for each tissue and cell type from our final evaluations of the entire system (Figs. 3, 5, 7, and 8), and values have been added to separate tables in the SI.

Based on this analysis, we can conclude that the devices do produce more reproducible results than controls. This was often the case when using the static operational mode. However, the interval mode produced the most precise results in all cases. The main text has been edited accordingly for each tissue type. We again apologize for our oversight, and sincerely thank the reviewer for continuing to help us connect with this important metric.

Reviewer #2

The authors' response to the major criticism that this is simply an incremental advance over the published series of papers by the same authors is not satisfactory. The authors claim a lot of "soft" operational optimization advances but nothing fundamentally important to report to a scientific community. The technology aspect, albeit it might have required some tweaking to integrate the 3 components, is still identical to the published work. Moreover, the integration was not create a single chip to truly integrate all functions but rather it was done by using standard tubes and lines to connect 3 devices together. The biological aspect is standard single cell sequencing and no new

information is generated. It is also not clear what it means the main purpose of the paper is to contact with end-users. This paper might be more appropriate for a trade journal or a company advertisement but not a scientific paper.

We respectfully disagree with this assessment. We stand by our points from the first response, adding only that the primary scientific benefit of this study is to show that our platform can extract more single cells from a variety of tissues, and with less biasing of results obtained via scRNA-seq. Many in the scientific community will be interested in these capabilities, and we hope this study will help us forge important collaborations so that we can begin to generate new information about different cell types and functions in our future work. In addition, to better illustrate our integrated system, we have now added pictures and schematics to the SI.

Reviewer #3

My concerns have been addressed in the revised version.

Many thanks for the constructive comments.

Reviewers' Comments:

Reviewer #1:

Remarks to the Author:

In my opinion, the authors have satisfactorily responded to my remaining comment on reproducibility.